# Entropy-based Activation Function Optimization: A Method on Searching Better Activation Functions

**Haoyuan Sun**[1*], **Zihao Wu**[2*], **Bo Xia**[1], **Pu Chang**[3], **Zibin Dong**[2], **Yifu Yuan**[2],
**Yongzhe Chang**[1†], **Xueqian Wang**[1†]
[1]Tsinghua University   [2]Tianjin University   [3]Anhui Polytechnic University

## Abstract

The success of artificial neural networks (ANNs) hinges greatly on the judicious selection of an activation function, introducing non-linearity into network and enabling them to model sophisticated relationships in data. However, the search of activation functions has largely relied on empirical knowledge in the past, lacking theoretical guidance, which has hindered the identification of more effective activation functions. In this work, we offer a proper solution to such issue. Firstly, we theoretically demonstrate the existence of the **W**orst **A**ctivation **F**unction with **B**oundary **C**onditions (WAFBC) from the perspective of information entropy. Furthermore, inspired by the Taylor expansion form of information entropy functional, we propose the **E**ntropy-based **A**ctivation **F**unction **O**ptimization (EAFO) methodology. EAFO methodology presents a novel perspective for designing static activation functions in deep neural networks and the potential of dynamically optimizing activation during iterative training. Utilizing EAFO methodology, we derive a novel activation function from ReLU, known as **C**orrection **R**egularized **ReLU** (CRReLU). Experiments conducted with vision transformer and its variants on CIFAR-10, CIFAR-100 and ImageNet-1K datasets demonstrate the superiority of CRReLU over existing corrections of ReLU. Extensive empirical studies on task of large language model (LLM) fine-tuning, CRReLU exhibits superior performance compared to GELU, suggesting its broader potential for practical applications.

## 1 Introduction

Flourishing development of artificial intelligence is predominantly attributable to rapid advancements in artificial neural networks (ANNs) observed in recent years. Activation functions (AFs) play a critical role in the performance of ANNs due to their fundamental role in enabling nonlinear representations. Despite continuous development of novel activation functions and their empirical success in improving network performance, theoretical analysis towards these activation functions remain scarce in the research literature. In other words, proposal of improved activation functions is often based on empirical evidence without theoretical validations, which greatly hinders the search for better activation functions. Hence, a theoretically reliable methodology on searching better activation functions holds significant value for the machine learning community and future research.

> ── **Major Questions** ──
> 1. *Is there the best activation function?*
> 2. *If the answer is no, can we give a methodology for searching better activation functions?*
> 3. *Can we provide an example to illustrate how the given methodology is applied in practice?*

Our research endeavors are centered on addressing the above three questions. In our work, we initiate our exploration from the correlation between information entropy and Bayesian error rate; subsequently, we establish the connection between activation function and information entropy. To answer the first question, we derive the specific form that the *worst* activation function exists under boundary conditions and draw the conclusion that *there is no best activation functions but only better*

*Equal contribution: sun-hy23@mails.tsinghua.edu.cn, 3020210034@tju.edu.cn

†Corresponding authors: changyongzhe@sz.tsinghua.edu.cn, wang.xq@sz.tsinghua.edu.cn

*ones*. To answer the second question, we propose a novel method for optimizing activation functions, namely *the Entropy-based Activation Function Optimization (EAFO)* methodology. To answer the third question, we derive a novel activation function known as ***Correction Regularized ReLU (CRReLU)*** with the beginning of conventional ReLU utilizing the EAFO methodology. Experiments involving the vision transformer (Dosovitskiy et al., 2020) and its variants (Touvron et al., 2021; Han et al., 2021), conducted on CIFAR-10, CIFAR-100 (Krizhevsky et al., 2009) and ImageNet-1K (Deng et al., 2009) datasets, have consistently demonstrated the superior performance of CRReLU compared to other activation function baselines. Extensive experimental studies on the task of large language model (LLM) fine-tuning with direct preference optimization (DPO) method (Rafailov et al., 2023) also demonstrate that CRReLU surpasses GELU in performance, suggesting the wider applicability of CRReLU in practical scenarios. Moreover, the EAFO methodology also shows potential to further optimize activation functions during the iterative training of ANNs, although the specific optimization techniques remain a topic of ongoing research.

In summary, our main contributions are as follows:

- **We answer the first question: there is no best activation function, but better ones.** We theoretically prove the existence of the Worst Activation Function with Boundary Conditions (WAFBC) from the perspective of information entropy; and starting from the worst activation function, performance of activation functions always improves.
- **We answer the second question: the Entropy-based Activation Function Optimization (EAFO) methodology is proposed**, which provides a novel perspective for designing static activation functions in deep neural networks and the potential of dynamically optimizing activation during iterative training.
- **We answer the third question: a novel activation function is derived, known as Correction Regularized ReLU (CRReLU), which starts from the ReLU utilizing the EAFO methodology.** Experiments across mainstream architectures, datasets and tasks demonstrate that the proposed CRReLU exceeds existing activation functions, exhibiting exceptional performance.

## 2 RELATED WORK

With the development of deep learning, deep neural networks (DNNs) have gained significant prominence and achieved notable success across various domains. Recent advancements in the field of natural language processing, exemplified by large language models such as GPT-4 (OpenAI, 2023), LLama-3 (Hugo Touvron, 2023), and Gemini (Team, 2024), have propelled machine understanding and generation of natural language to unprecedented levels of accuracy. Additionally, deep neural networks have also achieved important applications in computer vision (Dosovitskiy et al., 2020; Touvron et al., 2021; Han et al., 2021), deep reinforcement learning (Schulman et al., 2017), autonomous driving(Pan et al., 2024), and many other areas.

The nonlinearity of activation functions in neural networks is crucial for both enabling the efficient learning of complex patterns, and facilitating the extraction of intricate and hierarchical representations from input data, thus allowing them to capture more complex relationships between input and output variables. In contrast, however, the nonlinear activation functions of deep neural networks also presents challenges during training, encompassing challenges like gradient vanishing (Bengio et al., 1994), gradient exploding (Larochelle et al., 2009), and so on.

To address these challenges, researchers have explored alternative approaches for improvement, including the enhancement of activation functions. In the nascent stages of activation function development, scholars predominantly focused on rudimentary thresholding functions, initially directing their attention towards squashing functions such as the Sigmoid function and the Tanh function (Hornik, 1991). In order to mitigate the issues of vanishing and exploding gradients, various non-squashing functions have been proposed. Notably, ReLU (Hahnloser et al., 2000; Jarrett et al., 2009; Nair & Hinton, 2010) has played a pivotal role in the remarkable success of deep learning. The derivative of ReLU for positive inputs is one, thereby preventing the gradient from vanishing; however, negative values are mapped to zero, leading to two main issues: (1) The absence of information flow for negative values, known as dying ReLU ; (2) The shift in subsequent layers due to positive bias maintained by activation.

Given the aforementioned challenges, researchers have dedicated significant efforts to improving the effectiveness of activation functions. The Leaky ReLU (Maas, 2013) activation function permits a small negative slope, ensuring some gradient can still be propagated even when input is less than zero. The Parametric ReLU (PReLU) (He et al., 2015) is an extension of the Leaky ReLU, where $\alpha$ is considered a learnable parameter that is learned from data rather than being predetermined. The Exponential Linear Unit (ELU) (Clevert et al., 2016) outputs a negative value when $x$ is less than 0, leading to the advantageous property of the average output approaching 0. The Continuously Differentiable Exponential Linear Unit (CELU) (Barron, 2017) proposes an alternative parameterization that simplifies analysis of the rectifier function and facilitates the tuning process of parameters in ELU. The Swish (also known as SiLU) (Ramachandran et al., 2017) has been shown to enhance training stability and performance in deep learning models due to its smooth nature and improved gradient propagation. In Mish (Misra, 2020) activation function, unboundedness of positive values avoids the saturation led by a plateau, slight allowance for negative values enables better gradient flow, and the smoother activation function allows better information to flow deep into neural networks, thus resulting in better accuracy and generalization in performance.

## 3 MOTIVATION

In Section 2, it is shown that researchers have dedicated substantial efforts to the exploration of improved activation functions, which are widely acknowledged to hold considerable significance for the advancement of deep learning. However, it has also come to our attention that proposals for these activation functions lack a theoretical framework, indicating such searches are, to some extent, inefficient and aimless.

GELU(Gaussian Error Linear Unit) (Hendrycks & Gimpel, 2023) was first proposed in 2016 and has since gained significant success in a variety of fields, especially with the emergence of large language models in recent years. It has been successfully incorporated into several cutting-edge neural network architectures, such as BERT(Devlin et al., 2019), ViT (Dosovitskiy et al., 2020), GPT-4(OpenAI, 2023), and so on, demonstrating its versatility and effectiveness. In the work conducted by Lee (2023), insightful mathematical properties of the GELU are finally unveiled, including its differentiability, boundedness, stationarity, and smoothness. Hence, it is often the case that superior performance exhibited by novel activation functions frequently lacks mathematical explanations for their observed enhancements. Understanding may merely limited to the fact that it exhibits improved performance, which hampers exploration for better activation functions and interpretability of neural networks.

In light of the aforementioned challenges, our work endeavors to propose a methodology for searching better activation functions, not only enabling the discovery of improved activation functions but also elucidating the reasons behind their superior performance at the same time.

## 4 METHODOLOGY

### 4.1 PROBLEM SETUP

#### 4.1.1 BAYESIAN ERROR RATE AND INFORMATION ENTROPY

A deep neural network can be simplified as comprising a feature extraction layer, which is subsequently followed by a fully connected layer for final classification. From a probabilistic perspective, in binary classification, the feature extraction layer can be conceptualized as transforming the shape of mixture distribution, thereby enabling the final fully connected layer to separate two distributions with a hyperplane. Hence, the more overlapping two distributions are, the higher Bayesian error rate and the worse classification performance. Furthermore, a lower information entropy corresponds to a higher likelihood of forming two distinct peaks (i.e. the smaller classification uncertainty, the easier to classify); and an increase in the overlap between two distributions also leads to the increase of information entropy (i.e. the greater classification uncertainty, the harder to classify). In addition, the above statements can be extended to multi-class classification, and further elaboration is omitted here. To better understand the statement that lower entropy indicates better classification, we conduct further discussion in Appendix I.

### 4.1.2 Activation Function and Information Entropy

Assuming the inverse function of the activation function is $y(x)$, and the activation function is monotonically increasing. Many previous activation functions, such as Sigmoid and Tanh (Hornik, 1991), satisfy the assumption that the function has an inverse function in entire definition domain. Furthermore, when an activation function fails to meet the assumption, we can transform the part of such function satisfying this assumption, as is the case with the positive part of ReLU.

Then we set data distribution before passing through the activation function obeys the distribution $p(x)$. Thus, data distribution after passing through activation function is : $q(x) = p(y(x)) y'(x)$, where $y'(x)$ represents the derivative of $y(x)$. Hence, we can express the information entropy as:

$$\mathbb{H}(y(x)) = -\int q(x) \log q(x) \mathrm{d}x = -\int p(y(x)) y'(x) \log(p(y(x)) y'(x)) \mathrm{d}x = \int \mathbb{G}(y'(x), y(x)) \mathrm{d}x$$

Therefore, the information entropy can be deemed as a **functional**, which takes a function $y(x)$ as input and produces a real number as output.

### 4.2 Worst Activation Function with Boundary Condition (WAFBC)

> **Answer to Question 1**: There is no best activation function, but only better ones. Furthermore, there exists the worst activation function, namely the **W**orst **A**ctivation **F**unction with **B**oundary **C**ondition (WAFBC).

Firstly, we would like to determine the extremum (whether it is a maximum or minimum) of the functional $\mathbb{H}(y(x))$. For further deductions, taking the simplest functional into consideration, e.g. setting $\mathbb{H}(y(x)) = \int \mathbb{G}(y'(x), y(x), x) \mathrm{d}x$.

In order to research the influence brought by variations of function $y(x)$, we apply a small perturbation $\varepsilon \eta(x)$ to function $y(x)$, and then the functional $\mathbb{H}(y(x) + \varepsilon \eta(x))$ takes the form as:

$$\mathbb{H}(y(x) + \varepsilon \eta(x)) = \int \mathbb{G}(y'(x) + \varepsilon \eta'(x), y(x) + \varepsilon \eta(x), x) \mathrm{d}x$$

We apply **Taylor expansion** to functional $\mathbb{H}(y(x) + \varepsilon \eta(x))$, we can obtain the following equation:

$$
\begin{aligned}
\mathbb{H}(y(x) + \varepsilon \eta(x)) &= \int \left[ \mathbb{G}(y'(x), y(x), x) + \varepsilon \frac{\partial \mathbb{G}}{\partial y'} \eta'(x) + \varepsilon \frac{\partial \mathbb{G}}{\partial y} \eta(x) + \mathcal{O}(\varepsilon) \right] \mathrm{d}x \\
&= \mathbb{H}(y(x)) + \varepsilon \int \left[ \frac{\partial \mathbb{G}}{\partial y} \eta(x) + \frac{\partial \mathbb{G}}{\partial y'} \eta'(x) \right] \mathrm{d}x + \mathcal{O}(\varepsilon)
\end{aligned}
\tag{1}
$$

As illustrated in Section 4.1.2, $q(x) = p(y(x)) y'(x)$ is the data distribution after passing through activation function. We can easily get that for the inverse function $y(x)$ of activation function, when $x$ approaches the lower bound (e.g. the initial activation function value approaches lower bound), $y(x)$ should approaches negative infinity; and when $x$ approaches the upper bound (e.g. the initial activation function value approaches upper bound), $y(x)$ should approaches positive infinity. And for the sake that $\varepsilon \eta(x)$ is a small perturbation applied to $y(x)$, we can draw the conclusion that $\eta(x)$ must be 0 at the boundaries.

Utilizing the method of integration by parts and boundary condition towards Equation 1, we can derive the following results:

$$\int \frac{\partial \mathbb{G}}{\partial y'} \eta'(x) \mathrm{d}x = \int \frac{\partial \mathbb{G}}{\partial y'} \mathrm{d}\eta(x) = \eta(x) \frac{\partial \mathbb{G}}{\partial y'} \Big|_x - \int \eta(x) \frac{\mathrm{d}}{\mathrm{d}x} \left( \frac{\partial \mathbb{G}}{\partial y'} \right) \mathrm{d}x = -\int \eta(x) \frac{\mathrm{d}}{\mathrm{d}x} \left( \frac{\partial \mathbb{G}}{\partial y'} \right) \mathrm{d}x$$

Thus, $\mathbb{H}(y(x) + \varepsilon \eta(x))$ has the following expression:

$$\mathbb{H}(y(x) + \varepsilon \eta(x)) = \mathbb{H}(y(x)) + \varepsilon \int \left[ \frac{\partial \mathbb{G}}{\partial y} - \frac{\mathrm{d}}{\mathrm{d}x} \left( \frac{\partial \mathbb{G}}{\partial y'} \right) \right] \eta(x) \mathrm{d}x + \mathcal{O}(\varepsilon)$$

In analogy to the extremum of ordinary functions, it is expected that the first-order term should be 0 at the extremum point. Such requirement for arbitrary $\eta(x)$ leads to the Euler-Lagrange equation:

$$\frac{\mathrm{d}}{\mathrm{d}x} \left( \frac{\partial \mathbb{G}}{\partial y'} \right) - \frac{\partial \mathbb{G}}{\partial y} = 0 \tag{2}$$

**Proposition 1.** *If $\mathbb{G}$ is independent of x, i.e. $\mathbb{G} = \mathbb{G}(y, y')$, based on the Euler-Lagrange equation expressed in Equation 2, then we have:*

$$\mathbb{G} - y' \frac{\partial \mathbb{G}}{\partial y'} = C \tag{3}$$

Detailed proof of Proposition 1 can be seen in Appendix A.

Substitute $\mathbb{G} = -p(y(x))y'(x)\log(p(y(x))y'(x))$ into Equation 3 and perform the calculation, the final result is:

$$\frac{dy}{dx}p(y(x)) = C$$

Integrating both sides of the equation simultaneously, the final solution is:

$$x = c_1 \int p(y)dy + c_2 \tag{4}$$

Based on the solution we get in Equation 4, for the sake that $y(x)$ is the inverse function of the activation function, the first integral equation can finally be solved to obtain the form of the activation function as:

$$f(x) = C_1 \int_{-\infty}^{x} p(t)dt + C_2, \tag{5}$$

where $C_1$ and $C_2$ are two constants based on the upper bound and lower bound of activation function. Equation 5 shows the analytical form of the worst activation function with boundary condition. We provide further discussion on this form in Appendix B. Through the above derivation, extremum of the functional is determined. Furthermore, we would like to deduce whether it is a maximum value or a minimum one. Applying **Legendre condition** to the functional extremum, then we have:

$$\mathbb{G}_{y'y'} = -\frac{p(y(x))}{y'} \leqslant 0$$

Therefore, the derived extremum is a maximum extremum, and is a global maximum extremum actually, meaning the deduced activation function has the worst performance. However, the minimum of the functional is needed if we would like to obtain the best activation function. Nonetheless, based on calculation, the actual situation is that this functional only has a global maximum but no global minimum exists. *Hence, there is no best activation function, but only better activation functions.* In this scenario, WAFBC represents a global maximum of the functional, indicating that the performance of activation functions consistently improves from WAFBC to any alternative activation functions.

### 4.3 ENTROPY-BASED ACTIVATION FUNCTION OPTIMIZATION (EAFO)

> **Answer to Question 2**: Entropy-based Activation Function Optimization (EAFO) methodology is proposed, which provides a novel perspective for designing static activation functions in deep neural networks and the potential of dynamically optimizing activation during iterative training.

Let's reconsider the Taylor expansion of the functional

$$\mathbb{H}(y(x) + \varepsilon\eta(x)) = \mathbb{H}(y(x)) + \varepsilon \int \left[ \frac{\partial \mathbb{G}}{\partial y} - \frac{d}{dx}\left(\frac{\partial \mathbb{G}}{\partial y'}\right) \right] \eta(x)dx + \mathcal{O}(\varepsilon)$$

To minimize the information entropy of novel activation function, it is advisable to reduce the first-order term of Taylor expansion. In order to ensure that the information entropy of novel activation function has been indeed reduced, we would like to set $\eta(x)$ as the opposite sign to $\frac{\partial \mathbb{G}}{\partial y} - \frac{d}{dx}\left(\frac{\partial \mathbb{G}}{\partial y'}\right)$, which means we set:

$$\eta(x) = -\left(\frac{\partial \mathbb{G}}{\partial y} - \frac{d}{dx}\left(\frac{\partial \mathbb{G}}{\partial y'}\right)\right) \tag{6}$$

Substitute the analytical form of functional $\mathbb{G}(y'(x), y(x))$ into Equation 6, perform the calculation, we can derive the following equation:

$$\eta(x) = -\left(p(y(x))\frac{y''(x)}{y'(x)} + p'(y(x))y'(x)\right), \tag{7}$$

where $p(x)$ is the probability density function (PDF) of data distribution before passing through the activation function; $p'(x)$ is the first order derivative of PDF; $y(x)$ is inverse function of the activation function; $y'(x)$ is the first order derivative of $y(x)$; $y''(x)$ is the second order derivative of $y(x)$.

As a result, we have derived a correction term that is capable of decreasing information entropy, expressing its general form in Equation 7. Subsequently, we can obtain the inverse function of the optimized activation function, denoted as $g(x) = y(x) + \eta(x)$. Finally, the optimized activation function can be obtained by deriving the inverse function of $g(x)$.

In summary, we express the theoretical EAFO methodology as follows:

> **EAFO methodology outline**
>
> **1)** Utilize Equation 7 and derive correction term $\eta(x)$ given data distribution $p(y)$ and inverse function of activation function $y(x)$.
> **2)** Sum the correction term with the inverse function to obtain the inverse function of the optimized function, i.e. $g(x) = y(x) + \eta(x)$.
> **3)** Derive the rigorous or approximate inverse function of $g(x)$, yielding the optimized activation function.

Furthermore, EAFO methodology has also shown the potential of dynamically optimizing activation during iterative training. We are acknowledged that activation of neural networks with Multi-Layer Perceptrons (MLPs) architecture is typically fixed. Recent studies, such as work done by Liu et al. (2024), have suggested the optimization of activation in innovative network architectures (such as Kolmogorov-Arnold Networks). Furthermore, across true data distributions $p(y)$, utilizing EAFO methodology, we may continuously optimize activation $y(x)$ practically under Multi-Layer Perceptrons (MLPs) architecture with numerical methods. Moreover, in theory, it is feasible to optimize activation functions using methods such as gradient descent optimization of the information entropy functional; however, we are also aware that this would result in an explosion of computational complexity in large neural networks, which calls for practically efficient algorithms. Hence, at the present stage, the EAFO methodology is still in the theoretical stage, providing guidance for calculating the analytical form of better activation functions.

### 4.4 CORRECTION REGULARIZED RELU (CRRELU) : FROM RELU TO BETTER

> **Answer to Question 3**: A novel activation function is derived, known as **C**orrection **R**egularized **R**e**LU** (CRReLU), which starts from ReLU utilizing the EAFO methodology.

As illustrated in Section 4.2, it is theoretically true that the worst activation function exists, and we can determine its exact form. Actually, beginning with the worst activation function, the value of the functional $\mathbb{G}$ consistently decreases, indicating an improvement in the performance of activation function. This reveals the feasibility of searching an improved activation function, which constitutes the crux of "optimization". In Section 4.3, EAFO is proposed as the optimization methodology. Hence, we can easily think of optimizing from WAFBC to get a better-performing activation function. While it is true that such an idea is feasible, we also observe that WAFBC itself takes the form of a variable upper bound integral, which yields a complex form of $\eta(x)$ and renders the deduced result not practically significant. Moreover, commencing optimization from WAFBC also leads to sluggish advancement. Therefore, from the practical perspective, we are inclined to start from an activation function that already demonstrates relatively good performance.

Here, we would like to take ReLU (Hahnloser et al., 2000; Jarrett et al., 2009; Nair & Hinton, 2010) as the beginning, and show the process of finding a better activation function. Before the deduction, we also notice that ReLU is lack of an inverse function over the entire domain. In this section, we would like to utilize following strategies for mitigating the aforementioned dilemma: the initial activation function only necessitates an inverse function in specific regions where it is required; and when encountering parts without an inverse function, we may employ practical approximations. Therefore, we initially examine the region where $x$ is positive in the case of ReLU. As shown in Equation 7, the derivation of correction term $\eta(x)$ only requires original distribution $p(y)$ and inverse function of the activation function $y(x)$. Knowledge of activation function is easily available, whereas original distribution remains unexplored. However, in real experiments, original distribution of experimental

data would surely exhibit a substantial degree of morphological variability, thus lacking a perfect analytical form. Hence, we assume the situation is that networks are large enough, according to the Central Limit Theorem, the data processed by them can be approximated as a Gaussian distribution in MLPs (Williams, 1996; Lee et al., 2018; Park et al., 2020; Gao et al., 2023) and CNNs (Huang et al., 2021). Certainly, such assumption may not always hold in networks of real experiments and modern architectures (such as Tranformers). Nevertheless, approximation of the exact solution for inverse function and introduction of the learnable parameter $\varepsilon$ could have significantly mitigated the impact of such assumption, which can also be demonstrated by the insensitivity of CRReLU to data distribution shown in Section 5.

Now, let's consider the derivation from ReLU to CRReLU. For the sake of concise representation, we rewrite the data distribution and the derivative of data distribution as:

$$p(y) = C \cdot e^{-\frac{y^2}{2}}, \; p'(y) = -C \cdot y e^{-\frac{y^2}{2}}$$

Furthermore, ReLU has a mathematical function defined as $y = x$ when $x$ is positive, meaning we have $y(x) = x$, $y'(x) = 1$ and $y''(x) = 0$. Therefore,

$$p'(y(x)) = p'(x) = -C \cdot y e^{-\frac{y^2}{2}} = -C \cdot x e^{-\frac{x^2}{2}}$$

Ultimately, incorporating $p'(y) = -C \cdot x e^{-\frac{x^2}{2}}$, $y'(x) = 1$ and $y''(x) = 0$ into Equation 7, we can obtain:

$$\eta(x) = -C \cdot x e^{-\frac{x^2}{2}}$$

Furthermore, we *make constant C as a learnable parameter $\varepsilon$* with the purpose of enabling *self-optimization* in networks. According to EAFO methodology, we can get the inverse function of revised activation function as follows:

$$g(x) = x - \varepsilon x e^{-\frac{x^2}{2}} \qquad x \geqslant 0 \tag{8}$$

Finally, the optimized activation function CRReLU can be obtained by deriving the inverse function of $g(x)$. However, obtaining the inverse function of Equation 8 presents a challenge using conventional methods; as a consequence, we use the following function as a form of practical approximation.

$$f(x) = x + \varepsilon x e^{-\frac{x^2}{2}} \qquad x \geqslant 0 \tag{9}$$

We show the rationalization and reliability of utilizing Equation 9 as the approximate inverse function of Equation 8 in Proposition 2

**Proposition 2.** *Known $g(x) = x - \varepsilon x e^{-\frac{x^2}{2}}, f(x) = x + \varepsilon x e^{-\frac{x^2}{2}}$, for $x \geqslant 0$, the absolute value of error between $g(f(x))$ and $x$ is bounded with $\left| e^{-1} \varepsilon^2 + 0.5 e^{-\frac{3}{2}} \varepsilon^3 \right|$.*

Detailed proof of Proposition 2 can be seen in Appendix C.

As illustrated in Section 4.2, $\varepsilon \eta(x)$ is the small perturbation; hence, from a theoretical perspective, $\varepsilon \eta(x)$ can be set as an infinitesimal. Furthermore, given the knowledge that $\eta(x)$ is a bounded function, we can deduce that $\varepsilon$ is also an infinitesimal. Therefore, absolute value of error between $g(f(x))$ and $x$ is an infinitesimal of higher order. In practice, we typically initialize $\varepsilon$ to a small value, such as 0.01 (as described in Section 5), implying that the absolute value of error is a small value.

Finally, let's consider the part where $x$ is negative. When $x$ is negative, the inverse function of ReLU can be visualized as a ray emanating from the origin and extending to infinity, possessing an infinite slope; and when $x$ is positive, it constitutes a ray with the slope of 1. Hence, the correction term solution for both positive and negative values of $x$ can be considered identical, differing only by constant $C$. In Equation 9 and Proposition 2, it is shown that incorporating the correction term into a linear activation function can have beneficial effects by reducing the information entropy. Therefore, we can obtain the full form of Correction Regularized ReLU as:

$$f(x) = \max(0, x) + \varepsilon x e^{-\frac{x^2}{2}} \tag{10}$$

**Lipschitz Continuity Analysis of CRReLU.** Lipschitz continuity constitutes a stronger form of the continuity, which imposes an upper bound on the rate of variation of a function. We would like to begin with the defination of Lipschitz-continuous functions.

**Definition 1** (**Lipschitz Continuous Function**(Khromov & Singh, 2024)). *For function $f: \mathbb{R}^d \rightarrow \mathbb{R}^K$, defined on some domain $dom(f) \subseteq \mathbb{R}^d$, is called C-Lipschitz continuous, $C > 0$, w.r.t. some $\alpha$-norm if, $\forall x, y \in dom(f)$: $\|f(x) - f(y)\|_\alpha \leq C\|x - y\|_\alpha$.*

We are interested in the *smallest C* that makes the above condition hold. This is what called the *(true) Lipschitz constant of function f*. Lipschitz continuity of an activation function is crucial for ensuring a well-behaved optimization landscape, thereby promoting efficient convergence during training; furthermore, a Lipschitz constant closer to 1 indicates stronger Lipschitz continuity. Through derivation, we can calculate Lipschitz constant of GELU (Lee, 2023), SiLU, Mish and CRReLU.

**Fact 1.** *Lipschitz constant of GELU is 1.084; Lipschitz constant of SiLU is 1.100; Lipschitz constant of Mish is 1.089.*

**Fact 2.** *Under mild assumptions, Lipschitz constant of CRReLU is $\max(1 + \varepsilon, 1 - 0.446\varepsilon)$.*

Based on this, we further obtain the recommended initialization range for $\varepsilon$ of CRReLU in Corollary 1.

**Corollary 1.** *In order to make Lipschitz constant of CRReLU remains lower than that of GELU, the range of $\varepsilon$ is (-0.188,0.084). We recommend setting the initial value of $\varepsilon$ within this range.*

Detailed proof of Fact 1, Fact 2 and Corollary 1 can be seen in Appendix E. Moreover, we conduct further discussion on the initialization of $\varepsilon$ and training stability in Appendix H.

Finally, we provide further details of CRReLU in Appendix D, including python-like pseudocode of CRReLU in Appendix D.1, and further discussion on properties of CRReLU in Appendix D.2.

## 5 EXPERIMENTS

**Datasets.** In experiments of image classification task, we adopt three datasets, ordered as CIFAR-10, CIFAR-100 (Krizhevsky et al., 2009) and ImageNet-1K (Deng et al., 2009) in terms of the number of classification categories. In experiments of large language model (LLM) fine-tuning task, we employ two human preference datasets: SHP (Ethayarajh et al., 2022) and HH (Bai et al., 2022).

**Baselines.** We conduct experiments comparing the performance of CRReLU with several typical existing corrections of ReLU as illustrated in Section 2 and Section 3 : PReLU (He et al., 2015), ELU (Clevert et al., 2016), CELU (Barron, 2017), GELU (Hendrycks & Gimpel, 2023), Swish (SiLU) (Ramachandran et al., 2017) and Mish (Misra, 2020).

**Experimental hyperparameters.** For all transformer-based architectures, we directly set $\varepsilon$ to 0.01 without further optimization. Detailed experimental hyperparameters are provided in Appendix N.

### 5.1 TASK OF IMAGE CLASSIFICATION

We conduct all experiments within CIFAR10 and CIFAR100 on 4×RTX3090 and those within ImageNet1K on 4×NVIDIA L20 for 100 epochs using the AdamW optimizer with weight decay of 0.05, truncated normal initialization (we provide further discussion on the reason for abandoning pre-trained initialization method in Appendix J.1), gradient clipping norm of 1.0, cross entropy loss function, and cosine annealing learning rate scheduler with linear warm-up. All experiments are conducted three runs, we report the mean and standard derivation.

**Experiments of ViTs on CIFAR-10 and CIFAR-100.** Vision Transformer and its variants possess sufficiently complex structure and representational capability, garnering widespread attention from the community. Moreover, the assumption of Gaussian distribution has been theoretically proved as reasonable for sufficiently large MLPs and CNNs; however, the distribution of data under attention mechanism of transformers remains unexplored. Hence, we select vision transformer and its variants as our test model in order to further investigate the insensitivity of CRReLU to data distribution. Phase of experiments on CIFAR-10 and CIFAR-100 involves the selection of Vision Transformer (ViT) (Dosovitskiy et al., 2020), Data-Efficient Image Transformer (DeiT) (Touvron et al., 2021) and Transformer in Transformer (TNT) (Han et al., 2021). We report the top-one accuracy on CIFAR-10 in Table 1 and CIFAR-100 in Table 2, demonstrating CRReLU outperforms other existing corrections of ReLU on the CIFAR datasets.

**Experiments of ViTs on ImageNet-1K.** ImageNet-1K dataset poses a significant challenge to the information processing capability of neural networks due to its large image size and extensive range

Table 1: Test accuracy of experiments conducted on CIFAR-10 for 100 epochs.

| Top-one Accuracy | | GELU | ELU | PReLU | CELU | SiLU | Mish | **CRReLU** |
|---|---|---|---|---|---|---|---|---|
| CIFAR-10 | ViT-Tiny | 70.4±0.2 | 66.4±0.5 | 78.0±0.6 | 66.5± 0.6 | 68.6±0.3 | 68.7±0.3 | **80.7±0.3** |
| CIFAR-10 | DeiT-Tiny | 72.4±0.7 | 67.6±0.6 | 75.4±0.1 | 67.7±0.8 | 69.9±0.5 | 70.2±0.6 | **77.0±0.3** |
| CIFAR-10 | TNT-Small | 73.7±0.5 | 69.5±0.6 | 75.8±0.3 | 68.7±0.2 | 71.1±0.7 | 71.6±0.8 | **76.9±0.5** |

Table 2: Test accuracy of experiments conducted on CIFAR-100 for 100 epochs.

| Top-one Accuracy | | GELU | ELU | PReLU | CELU | SiLU | Mish | **CRReLU** |
|---|---|---|---|---|---|---|---|---|
| CIFAR-100 | ViT-Tiny | 32.6±0.8 | 28.9±0.1 | 43.2±1.0 | 28.9±0.2 | 31.2±0.6 | 30.6±0.8 | **46.6±0.6** |
| CIFAR-100 | DeiT-Tiny | 46.6±0.9 | 40.5±0.5 | 50.0±0.5 | 40.5±0.5 | 43.5±0.6 | 43.8±1.0 | **50.7±0.1** |
| CIFAR-100 | TNT-Small | 47.5±0.8 | 43.6±0.3 | 49.0±0.7 | 43.0±0.5 | 45.0±0.9 | 45.5±0.8 | **50.9±0.4** |

Table 3: Test accuracy of experiments conducted on ImageNet-1K for 100 epochs.

| Top-one Accuracy | | GELU | ELU | PReLU | CELU | SiLU | Mish | **CRReLU** |
|---|---|---|---|---|---|---|---|---|
| ImageNet-1K | ViT-Tiny | 53.9±0.3 | 37.2±0.6 | 56.8±0.3 | 37.6±0.5 | 46.1±0.7 | 46.9±1.1 | **57.5±0.4** |
| ImageNet-1K | DeiT-Tiny | **61.7±0.4** | 49.1±0.7 | 60.8±0.4 | 48.9±0.8 | 58.5±0.7 | 58.9±0.3 | **61.6±0.2** |

of classification categories. Hence, we further conduct experiments on ImageNet-1K with Vision Transformer (ViT) (Dosovitskiy et al., 2020) and Data-Efficient Image Transformer (DeiT) (Touvron et al., 2021). We report the top-one accuracy on ImageNet-1K in Table 3 (and we provide further discussion on the results of DeiT in Appendix J.2).

**Additional Experiments on Architecture and Dataset.** In Appendix F, we conduct experiments on ConvNeXt and EuroSAT to verify generalization of CRReLU to network architecture and dataset.

**Entropy Analysis across Network Layers.** In order to provide more insights on CRReLU's behavior, we would like to further compare the post-training entropy of neural networks trained with CRReLU and GELU. We apply post-trained ViT-Tiny with CRReLU and GELU on ImageNet1K. By randomly selecting same ten batches of images from ImageNet1K, we compute the information entropy after each of the 12 layers. We present the mean and standard deviation of the results in Table 4.

Table 4: Entropy calculation after activation on 12 layers of post-trained ViT-Tiny on ImageNet1K.

| Layer | 1 | 2 | 3 | 4 | 5 | 6 |
|---|---|---|---|---|---|---|
| CRReLU | 7.594±0.007 | 7.598±0.003 | 7.599±0.003 | 7.595±0.003 | 7.592±0.003 | 7.584±0.004 |
| GELU | 7.536±0.046 | 7.541±0.019 | 7.561±0.011 | 7.573±0.006 | 7.580±0.005 | 7.583±0.004 |

| Layer | 7 | 8 | 9 | 10 | 11 | 12 |
|---|---|---|---|---|---|---|
| CRReLU | 7.572±0.005 | 7.557±0.005 | 7.540±0.005 | 7.523±0.007 | 7.498±0.008 | 7.461±0.008 |
| GELU | 7.585±0.004 | 7.585±0.004 | 7.583±0.004 | 7.580±0.004 | 7.577±0.004 | 7.560±0.004 |

From the results presented above, it is shown that for GELU, the entropy after 12 layers of activation exhibits an overall increasing trend; conversely, CRReLU demonstrates a general declining trend. Furthermore, we note that the reduction in entropy for CRReLU between layers 1 and 6 is not significant, whereas a marked decline is observed from layers 7 to 12. Moreover, we conduct experiments under mixed activation function (6GELU+6CRReLU), with results presented in Appendix G.

## 5.2 Task of Large Language Model (LLM) Fine-tuning

In order to further validate the effectiveness of CRReLU on larger networks and generalization to a richer range of applications, we further perform supplementary experiments on LLM fine-tuning task. We employ the Direct Preference Optimization (DPO) (Rafailov et al., 2023) method to fine-tune GPT-2 (Radford et al., 2019) on Stanford Human Preferences (SHP) dataset (Ethayarajh et al., 2022) and Anthropic HH dataset (Bai et al., 2022). The parameter number of GPT-2 is 137 M, a relatively modest magnitude, hence we conduct full fine-tuning instead of LoRA-based one on 2×RTX3090. Firstly, we carry out supervised fine-tuning (SFT) with the purpose of mitigating distribution shift between the true reference distribution which is unavailable, and the reference policy utilized by DPO. Subsequently, we separately set the penalty coefficient $\beta$ as **0.1, 1, 2, and 5**, in order to compare the

Table 5: Metrics comparison between CRReLU and GELU in the task of LLM fine-tuning.

| Evaluation Metrics | | Evaluation Margin Reward↑ | Evaluation Accuracy↑ | Evaluation Loss↓ |
|---|---|---|---|---|
| $\beta = 0.1$ | CRReLU | **0.1428±0.0002** | **0.6209±0.0001** | **0.6476±0.0000** |
| | GELU | 0.1420±0.0003 | 0.6197±0.0001 | 0.6480±0.0000 |
| $\beta = 1$ | CRReLU | **0.4627±0.0007** | **0.5757±0.0001** | **0.9202±0.0002** |
| | GELU | 0.4560±0.0006 | 0.5729±0.0003 | 0.9387±0.0008 |
| $\beta = 2$ | CRReLU | **0.7757±0.0021** | **0.5631±0.0003** | **1.4610±0.0008** |
| | GELU | 0.7178±0.0015 | 0.5606±0.0001 | 1.4814±0.0005 |
| $\beta = 5$ | CRReLU | **1.8473±0.0032** | **0.5635±0.0002** | **3.2677±0.0006** |
| | GELU | 1.6538±0.0069 | 0.5568±0.0003 | 3.3034±0.0021 |

performance of CRReLU and GELU under different penalty coefficients, and then execute DPO. We report mean and standard deviation of evaluation metrics across the fine-tuning process in Table 5, which demonstrates that CRReLU generally outperforms GELU in LLM fine-tuning task.

## 6 DISCUSSION

Pursuit of better activation functions has been a longstanding and fundamental topic in the realm of machine learning. However, prior research has consistently concentrated on empirical search, without an emphasis on understanding the underlying mathematical mechanisms. This work aims to offer a proper solution to such issue. Our investigation into the relationship between activation functions and information theory concepts reveals that information entropy can be represented as a functional. Existence of the worst activation function with boundary condition (WAFBC) furnishes a solid theoretical basis for exploring better activation functions. In the process of solving WAFBC, we draw inspiration from the Taylor expansion form, leading us to propose Entropy-based Activation Function Optimization (EAFO) methodology. EAFO methodology presents a novel perspective for designing static activation functions in deep neural networks and shows the potential of dynamically optimizing activation during iterative training. Utilizing EAFO methodology, we derive a novel activation function from ReLU, called Correction Regularized ReLU (CRReLU). Experiments involving image classification task and large language model (LLM) fine-tuning task demonstrate that CRReLU is comparable to or surpasses existing corrections of ReLU. Overall, the EAFO methodology provides numerous promising avenues for future research on activation functions, and the CRReLU introduces a novel addition to the set of high-performing activation functions.

**Limitations and Future Work.** Our findings raise several important questions for future work. *Firstly*, how can EAFO framework be systematically generalized to non-invertible activation functions? In the initial setting of EAFO methodology, the choice of activation function is restricted to those with invertible counterparts. One potential approach to addressing this problem entails employing the Lebesgue integral form instead of the original Riemann integral utilized in the entropy calculation. *Secondly*, how to effectively implement activation function iteration optimization during neural network training? Notwithstanding the demonstrated feasibility of iterative activation function optimization during neural network training, it is currently hindered by the high computational complexity, particularly in large-scale neural networks. Applicability of the EAFO methodology to optimize activation in alternative network structures, such as Kolmogorov-Arnold Networks (KANs), also deserves further in-depth research. Therefore, the development of practical and efficient algorithms is an exciting direction for future work. We provide further discussion on dynamic optimization in Appendix K. *Moreover*, although previous work has theoretically established the reasonability of Gaussian distribution assumptions for MLPs and CNNs, and we have empirically demonstrated superiority of CRReLU in modern architectures like transformers, the theoretical robustness under a broader range of distributions is still worthy of further research. *Additionally*, since there have already been more than 400 activation functions, it is worth exploring in greater depth how to theoretically rank them; and we provide further discussion on activation function ranking in Appendix L. *Finally*, while we have empirically validated the exceptional performance of CRReLU on image classification task and large language model fine-tuning task, its performance on other tasks remains to be explored (we provide further discussion on large language model inference task in Appendix M), thereby warranting further investigation in the future work.

## ACKNOWLEDGMENTS

This work is partly supported by the National Natural Science Foundation of China (No.62103225), Natural Science Foundation of Shenzhen (No.JCYJ202308807111604008), Natural Science Foundation of Guangdong Province (No.2024A1515010003) and National Key Research and Development Program (No.2022YFB4701402).

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

## A    PROOF OF PROPOSITION 1

*Proof.* From Equation 2, we know that:

$$\frac{\mathrm{d}}{\mathrm{d}x}(\frac{\partial \mathbb{G}}{\partial y'}) - \frac{\partial \mathbb{G}}{\partial y} = 0$$

Considering the total differential of $\mathbb{G}$:

$$\frac{\mathrm{d}\mathbb{G}}{\mathrm{d}x}(y', y, x) = \frac{\partial \mathbb{G}}{\partial x} \cdot \frac{\mathrm{d}x}{\mathrm{d}x} + \frac{\partial \mathbb{G}}{\partial y} \cdot \frac{\mathrm{d}y}{\mathrm{d}x} + \frac{\partial \mathbb{G}}{\partial y'} \cdot \frac{\mathrm{d}y'}{\mathrm{d}x} = \frac{\partial \mathbb{G}}{\partial x} + \frac{\partial \mathbb{G}}{\partial y} \cdot y' + \frac{\partial \mathbb{G}}{\partial y'} \cdot y''$$

Thus, we have:

$$\begin{aligned}
\frac{\mathrm{d}}{\mathrm{d}x}\left(y'\frac{\partial \mathbb{G}}{\partial y'}\right) &= y''\frac{\partial \mathbb{G}}{\partial y'} + y'\frac{\mathrm{d}}{\mathrm{d}x}\left(\frac{\partial \mathbb{G}}{\partial y'}\right) \\
&= \frac{\mathrm{d}\mathbb{G}}{\mathrm{d}x}(y', y, x) - \frac{\partial \mathbb{G}}{\partial y} \cdot y' - \frac{\partial \mathbb{G}}{\partial x} + y'\frac{\mathrm{d}}{\mathrm{d}x}\left(\frac{\partial \mathbb{G}}{\partial y'}\right) \\
&= \frac{\mathrm{d}}{\mathrm{d}x}\mathbb{G}(y', y, x) - \frac{\partial \mathbb{G}}{\partial x} - y' \cdot \left(\frac{\partial \mathbb{G}}{\partial y} - \frac{\mathrm{d}}{\mathrm{d}x}\left(\frac{\partial \mathbb{G}}{\partial y'}\right)\right) \\
&= \frac{\mathrm{d}}{\mathrm{d}x}\mathbb{G}(y', y, x) - \frac{\partial \mathbb{G}}{\partial x}
\end{aligned}$$

Therefore, we know that

$$\frac{\partial \mathbb{G}}{\partial x} - \frac{\mathrm{d}}{\mathrm{d}x}\left(\mathbb{G} - y'\frac{\partial \mathbb{G}}{\partial y'}\right) = 0$$

For the sake that $\mathbb{G}$ is independent of $x$, then we have that $\frac{\partial \mathbb{G}}{\partial x} = 0$. Hence,

$$\frac{\mathrm{d}}{\mathrm{d}x}\left(\mathbb{G} - y'\frac{\partial \mathbb{G}}{\partial y'}\right) = 0$$

Finally, we can draw the conclusion that:

$$\mathbb{G} - y'\frac{\partial \mathbb{G}}{\partial y'} = C,$$

which completes the proof. □

## B    FURTHER DISCUSSION ON WAFBC

Let's take several typical boundary conditions into consideration. Firstly, setting $f(x)$ approaches 1, when $x$ tends to positive infinity; and $f(x)$ approaches 0, when $x$ tends to negative infinity. Therefore, the solution takes the form of cumulative distribution function (CDF), which can be expresses as:

$$f(x) = \int_{-\infty}^{x} p(t)\mathrm{d}t$$

Similarly, if fixing the difference between the upper and lower bounds of the activation function to be $e$, and making the activation function symmetric about the origin, the form can be written as:

$$f(x) = e\int_{0}^{x} p(t)\mathrm{d}t$$

Furthermore, in the event that the input data distribution is assumed to be approximately uniformly distributed, the worst activation function can be approximated as a linear function. Were it to approximate the input data distribution as a normal distribution, then the form of the worst activation function would be closer to Sigmoid and Tanh. Additionally, the WAFBC possesses some intriguing properties, for example, it inherently has upper and lower bounds, which can explain why bounded activation functions like Sigmoid and Tanh do not perform as well as unbounded functions like ReLU. We show the comparison of function curves in Figure 1 and Figure 2.

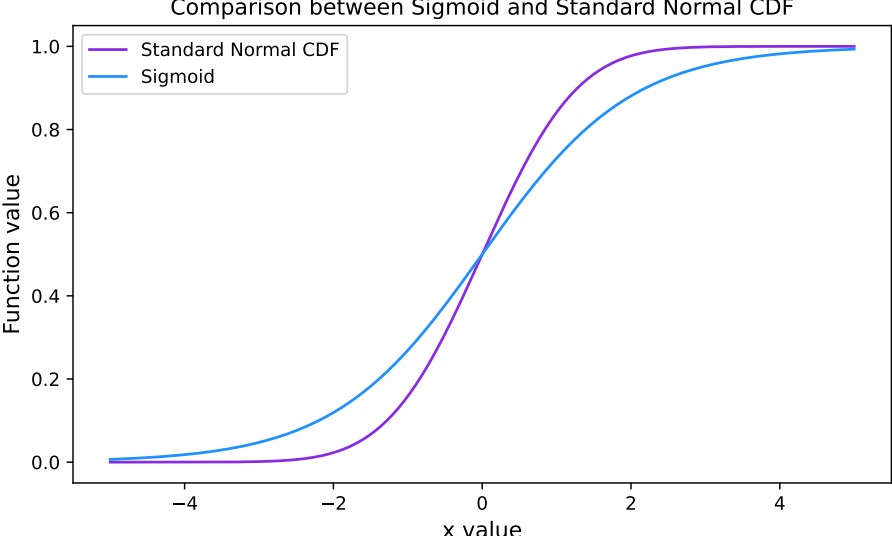

Figure 1: Comparison between Sigmoid and standard normal CDF

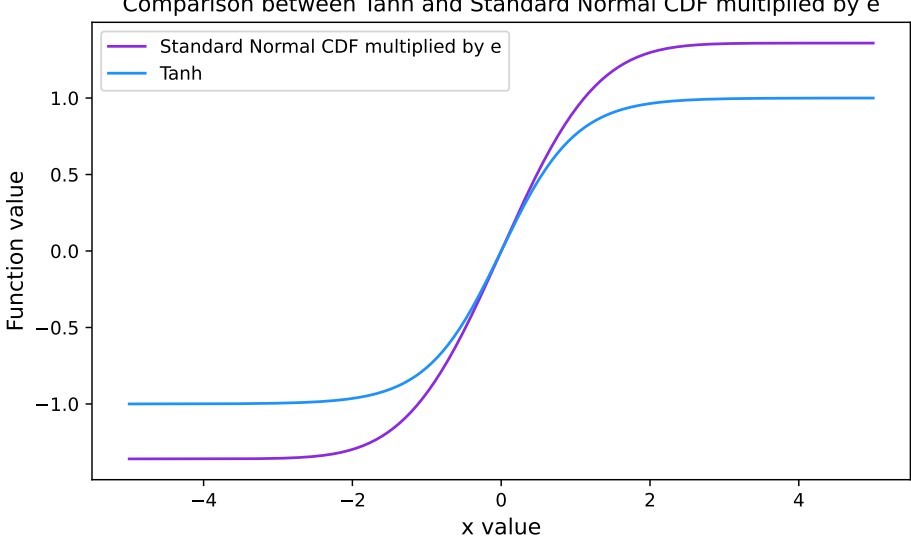

Figure 2: Comparison between Tanh and Standard Normal CDF multiplied by $e$ (has been transformed to achieve symmetry about origin)

## C  PROOF OF PROPOSITION 2

Before the proof of Proposition 2, we would like to show four facts without proof.

**Fact 3.** $f(x) = xe^{-\frac{x^2}{2}}$ is a bounded function, and range of the function is $[-e^{-\frac{1}{2}}, e^{-\frac{1}{2}}]$.

**Fact 4.** $f(x) = x^2 e^{-x^2}$ is a bounded function, and range of the function is $[0, e^{-1}]$.

**Fact 5.** $f(x) = x^3 e^{-\frac{3}{2}x^2}$ is a bounded function, and range of the function is $[-e^{-\frac{3}{2}}, e^{-\frac{3}{2}}]$.

**Fact 6.** $\forall x \in \mathcal{R}, 1 - e^{-x} - x \leqslant 0$.

We now commence the proof of Proposition 2.

*Proof.* Substituting the analytic expression into the formula and performing algebraic simplifications, we can obtain:

$$g\left(f(x)\right) = g\left(x + \varepsilon x e^{-\frac{x^2}{2}}\right) = x + \varepsilon x e^{-\frac{x^2}{2}} - \varepsilon\left(x + \varepsilon x e^{-\frac{x^2}{2}}\right) e^{-\frac{1}{2}\left(x + \varepsilon x e^{-\frac{x^2}{2}}\right)^2}$$

$$= x + \varepsilon x \left(e^{-\frac{x^2}{2}} - e^{-\frac{1}{2}\left(x + \varepsilon x e^{-\frac{x^2}{2}}\right)^2}\right) - \varepsilon^2 x e^{-\frac{x^2}{2}} e^{-\frac{1}{2}\left(x + \varepsilon x e^{-\frac{x^2}{2}}\right)^2}$$

$$= x + \varepsilon x e^{-\frac{x^2}{2}}\left[1 - e^{-\frac{1}{2}\left(2\varepsilon x e^{-\frac{x^2}{2}} + \varepsilon^2 x^2 e^{-x^2}\right)}\right] - \varepsilon^2 x e^{-\frac{x^2}{2}} e^{-\frac{1}{2}\left(x + \varepsilon x e^{-\frac{x^2}{2}}\right)^2}$$

Thus,

$$\left|g(f(x)) - x\right| = \left|\varepsilon x e^{-\frac{x^2}{2}}\left[1 - e^{-\frac{1}{2}\left(2\varepsilon x e^{-\frac{x^2}{2}} + \varepsilon^2 x^2 e^{-x^2}\right)}\right] - \varepsilon^2 x e^{-\frac{x^2}{2}} e^{-\frac{1}{2}\left(x + \varepsilon x e^{-\frac{x^2}{2}}\right)^2}\right|$$

$$\leqslant \left|\varepsilon x e^{-\frac{x^2}{2}}\left[1 - e^{-\frac{1}{2}\left(2\varepsilon x e^{-\frac{x^2}{2}} + \varepsilon^2 x^2 e^{-x^2}\right)}\right]\right|$$

$$\leqslant \left|\varepsilon x e^{-\frac{x^2}{2}}\left[-\frac{2\varepsilon x e^{-\frac{x^2}{2}} + \varepsilon^2 x^2 e^{-x^2}}{2}\right]\right|$$

$$= \left|\varepsilon x e^{-\frac{x^2}{2}}\left(-\varepsilon x e^{-\frac{x^2}{2}} - \frac{1}{2}\varepsilon^2 x^2 e^{-x^2}\right)\right| = \left|\varepsilon^2 x^2 e^{-x^2} + \frac{1}{2}\varepsilon^3 x^3 e^{-\frac{3}{2}x^2}\right|$$

$$\leqslant \left|e^{-1}\varepsilon^2 + 0.5 e^{-\frac{3}{2}}\varepsilon^3\right|$$

The first inequality is established owing to Fact 3 and the fact that when $x$ is positive, the second term of absolute value must be positive. The second inequality is established owing to Fact 6. The third inequality is established owing to Fact 4 and Fact 5. Hence, we can draw the conclusion that the absolute value of error between $g\left(f(x)\right)$ and $x$ is bounded with $\left|e^{-1}\varepsilon^2 + 0.5 e^{-\frac{3}{2}}\varepsilon^3\right|$, which completes the proof. □

# D FURTHER DETAILS OF CRReLU

## D.1 CORRECTION REGULARIZED RELU (CRReLU) PSEUDOCODE

---

**Algorithm 1:** Correction Regularized ReLU (CRReLU) Pseudocode

---

```python
import torch
import torch.nn as nn
import torch.nn.functional as F

class CRReLU(nn.Module):
    def __init__(self,lr=0.01):
        super(CRReLU,self).__init__()
        self.lr = nn.Parameter(torch.tensor(lr))

    def forward(self,x):
        return F.relu(x)+self.lr*x*torch.exp(-x**2/2)
```

---

## D.2 Further Discussion on Properties of CRReLU

We show the function curves with different $\varepsilon$ values for CRReLU in Figure 3. As depicted in the figure, existence of the correction term in CRReLU brings several good properties. It allows propagation of gradient when input is less than zero, serving to alleviate the dying ReLU phenomenon to a certain degree; simultaneously, as $x$ approaches negative infinity, CRReLU also converges to 0, thereby guaranteeing sparsity of models in the negative part.

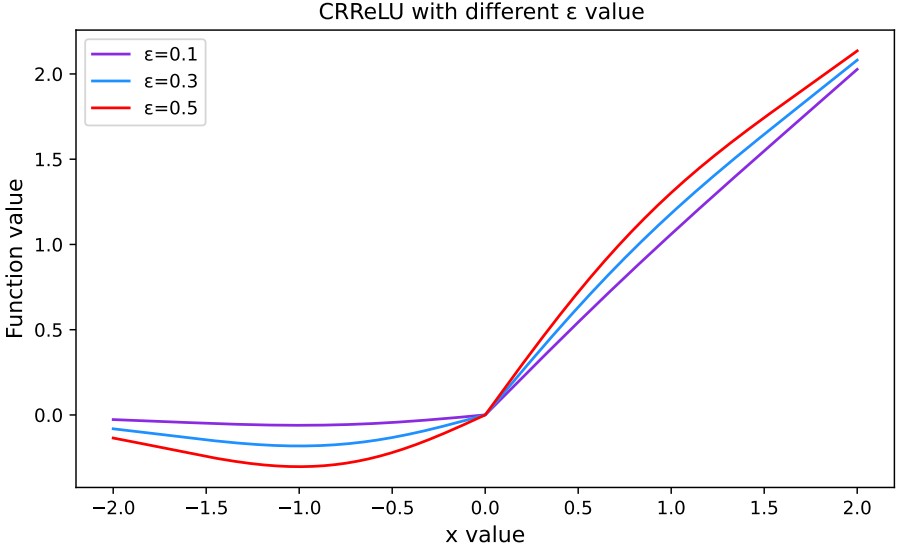

Figure 3: CRReLU with different $\varepsilon$ value

## E Proof of Lipschitz Continuity Analysis

**Remark 1** ((Lee, 2023)). *Lipschitz constant of GELU is 1.084.*

*Proof of sketch.* Firstly, compute the derivative of the GELU function as:
$$\frac{d\text{GELU}(x)}{dx} = x \cdot \frac{1}{\sqrt{2\pi}} e^{-\frac{x^2}{2}} + \Phi(x)$$

To find a constant C > 0 such that for all $x \in \mathbb{R}$, we have that $|\frac{d\text{GELU}(x)}{dx}| \leq C$.

Using the second derivative to establish a tight bound on the derivative. The second derivative of the GELU function is:
$$\frac{d^2\text{GELU}(x)}{dx^2} = \frac{1}{\sqrt{2\pi}} e^{-\frac{x^2}{2}} (2 - x^2)$$

Setting the second derivative equal to zero and solving for $x$, we obtain two critical points at $x = -\sqrt{2}$ and $x = \sqrt{2}$. Computing the first derivative at $x = \sqrt{2}$:
$$\frac{d\text{GELU}(x)}{dx}\Big|_{x=\sqrt{2}} \approx 1.084$$

Thus, it is found that derivation of GELU is bounded by 1.084, proving its Lipschitz continuity. $\square$

**Remark 2.** *Lipschitz constant of SiLU is 1.100.*

*Proof of sketch.* In the following calculations and derivations, we refer to the derivation in Remark 1 and make appropriate omissions. Firstly, we write the form of SiLU as :
$$\text{SiLU}(x) = \frac{x}{1 + e^{-x}}$$

We can compute the derivative of the SiLU function as:

$$\frac{d\text{SiLU}(x)}{dx} = \frac{(x+1)e^{-x}+1}{(1+e^{-x})^2}$$

and the second derivative as follows:

$$\frac{d^2\text{SiLU}(x)}{dx^2} = \frac{e^x(-e^x(x-2)+x+2)}{(1+e^x)^3}$$

We solve the transcendental equation $\frac{d^2\text{SiLU}(x)}{dx^2} = 0$ through a python program, we have $x_1 \approx -2.3994$ and $x_2 \approx 2.3994$. Furthermore, we have the derivative of the SiLU is bounded in the range of [-0.100, 1.100]. Hence, Lipschitz constant of SiLU is 1.100. $\qquad\square$

**Remark 3.** *Lipschitz constant of Mish is 1.089.*

*Proof of sketch.* Form of the Mish function, its derivation and its second derivation are performed as:

$$\text{Mish}(x) = x\frac{e^{2x}+2e^x}{e^{2x}+2e^x+2}$$

$$\frac{d\text{Mish}(x)}{dx} = \frac{e^x[4(x+1)+4e^{2x}+e^{3x}+e^x(4x+6)]}{(e^{2x}+2e^x+2)^2}$$

$$\frac{d^2\text{Mish}(x)}{dx^2} = \frac{4e^x(3e^{2x}(x-2)+2e^{3x}(x-1)-2(x+2)-2e^x(x+4))}{(e^{2x}+2e^x+2)^3}$$

Similarly, we solve the transcendental equation $\frac{d^2\text{Mish}(x)}{dx^2} = 0$ through a python program, we then have $x_1 \approx -2.2564$ and $x_2 \approx 1.4906$. Morever, the derivative of the Mish is bounded in the range of [-0.113, 1.089]. Hence, Lipschitz constant of Mish is 1.089. $\qquad\square$

**Remark 4.** *Under mild assumptions, Lipschitz constant of CRReLU is $\max(1+\varepsilon, 1-0.446\varepsilon)$.*

*Proof of sketch.*

$$\text{CRReLU}(x) = \begin{cases} x + \varepsilon x e^{-\frac{x^2}{2}} (x > 0) \\ \varepsilon x e^{-\frac{x^2}{2}} (x < 0) \end{cases}$$

Furthermore, under mild assumptions, we consider the derivative of CRReLU piecewise (disregarding the potential for non-differentiability at x = 0 temporarily).

$$\frac{d\text{CRReLU}(x)}{dx} = \begin{cases} 1 + \varepsilon(1-x^2)e^{-\frac{x^2}{2}} (x > 0) \\ \varepsilon(1-x^2)e^{-\frac{x^2}{2}} (x < 0) \end{cases}$$

Setting the second derviation to be 0

$$\frac{d^2\text{CRReLU}(x)}{dx^2} = \varepsilon e^{-\frac{x^2}{2}}(x^3 - 3x) = 0$$

Then we have that: $x_1 = 0, x_2 = \sqrt{3}, x_3 = -\sqrt{3}$. For $x_2$ and $x_3$, we directly calculate the values at that point:

$$\frac{d\text{CRReLU}(x)}{dx}\big|_{x_2} = 1 - 0.446\varepsilon; \quad \frac{d\text{CRReLU}(x)}{dx}\big|_{x_3} = -0.446\varepsilon$$

For $x_1 = 0$, we calculate the limit of $\frac{d\text{CRReLU}(x)}{dx}$ around 0:

$$\lim_{x\to0^+}\frac{d\text{CRReLU}(x)}{dx} = 1+\varepsilon; \lim_{x\to0^-}\frac{d\text{CRReLU}(x)}{dx} = \varepsilon$$

Hence, under mild assumptions, to obtain the upper bound for derivative of CRReLU. we have that:

$$C = \max(1+\varepsilon, \varepsilon, 1-0.446\varepsilon, -0.446\varepsilon)$$

Let's further consider the Lipschitz constant and simplify it. If $\varepsilon > 0$, it is obvious that $C = 1+\varepsilon$; and if $\varepsilon < 0$, we can get $C = 1 - 0.446\varepsilon$. Hence, we can express the Lipschitz constant of CRReLU as:

$$C = \max(1+\varepsilon, 1-0.446\varepsilon)$$

$\qquad\square$

**Corollary 1.** *In order to make Lipschitz constant of CRReLU remains lower than that of GELU, the range of $\varepsilon$ is (-0.188,0.084). We recommend setting the initial value of $\varepsilon$ within this range.*

*Proof of sketch.* If $\varepsilon > 0$, considering the following inequality:

$$1 + \varepsilon < 1.084, \text{ then we have: } 0 < \varepsilon < 0.084$$

If $\varepsilon < 0$, considering the following inequality:

$$1 - 0.446\varepsilon < 1.084, \text{ then we have: } -0.188 < \varepsilon < 0$$

□

**Corollary 2.** *If $\varepsilon$ is in (0.084,0.089)∪(-0.198, -0.188), we have the CRReLU's Lipschitz continuity worsen than GELU, but better than Mish.*

**Corollary 3.** *If $\varepsilon$ is in (0.089,0.100)∪(-0.224, -0.198), we have the CRReLU's Lipschitz continuity worsen than Mish, but better than SiLU.*

## F   ADDITIONAL EXPERIMENTS ON ARCHITECTURE AND DATASET

### F.1   ADDITIONAL ARCHITECTURE

In this section, we aim to enhance the evaluation of CRReLU's generalization to network architecture. We choose to validate performance on CIFAR10, CIFAR100, and ImageNet1K with the ConvNeXt-tiny (Liu et al., 2022). For all the experiments, we conduct three runs, and we report the mean and standard deviation in Table 6. Experiments within CIFAR10 and CIFAR100 are conducted on 4×RTX3090 and those within ImageNet1K are conducted on 4×NVIDIA L20.

Table 6: Test accuracy of experiments conducted on ConvNeXt-tiny for 100 epochs.

| Top-one Accuracy | | GELU | ELU | PReLU | CELU | SiLU | Mish | **CRReLU** |
|---|---|---|---|---|---|---|---|---|
| CIFAR10 | ConvNeXt | 64.9±0.4 | 59.8±0.5 | 64.6±1.4 | 59.8±0.5 | 60.6±0.2 | 61.4±0.4 | **70.6±1.1** |
| CIFAR100 | ConvNeXt | 36.6±0.3 | 30.3±0.4 | 35.2±0.5 | 30.5±0.2 | 35.0±0.9 | 35.3±0.7 | **42.1±0.7** |
| ImageNet1K | ConvNeXt | 72.9±0.3 | 71.7±0.5 | 72.9±0.5 | 71.8±0.9 | 72.3±0.7 | 72.8±0.6 | **73.2±0.2** |

### F.2   ADDITIONAL DATASET

In this section, we aim to enhance the performance of CRReLU on diverse datasets. We conduct experiments on EuroSAT (Helber et al., 2019) with ConvNeXt-tiny (Liu et al., 2022). All experiments are performed three times, we report the mean and standard deviation. We conduct this part experiments on a single RTX3090 for 25 epochs using the AdamW optimizer, learning rate of 0.0001, cross entropy loss function, batch size of 256.

Table 7: Test accuracy of experiments conducted with ConvNeXt-tiny on the EuroSAT.

| GELU | ELU | PReLU | CELU | SiLU | Mish | **CRReLU** |
|---|---|---|---|---|---|---|
| 83.09±1.06 | 81.21±0.37 | 81.33±0.94 | 81.12±0.27 | 81.85±1.01 | 82.23±0.08 | **83.26±0.52** |

## G   ADDITIONAL EXPERIMENTS ON MIXED ACTIVATION FUNCTION

In Table 4, it is shown that reduction in entropy for CRReLU between layers 1 and 6 is not significant. Hence, in this section, we focus on equipping different activation functions in the initial six layers and subsequent six ones. We employ GELU for layers 1 to 6 and CRReLU for layers 7 to 12, denoting this as "6GELU+6CRReLU". We conduct three runs on CIFAR10, CIFAR100, and ImageNet1K, presenting mean and standard deviation in Table 8. Experiments on CIFAR10 and CIFAR100 are conducted on 4×RTX3090, and those on ImageNet1K are carried out on 4×NVIDIA L20.

From the results, it appears that having only the last few layers equipped with CRReLU is not as effective as utilizing CRReLU throughout the entire network. Especially the results on ImageNet1K,

Table 8: Test accuracy of experiments conducted with ViT (12GELU, 6GELU+6CRReLU, 12CR-ReLU) for 100 epochs.

|  | 12GELU | 6GELU+6CRReLU | 12CRReLU |
|---|---|---|---|
| CIFAR10 | 0.704±0.002 | 0.755±0.008 | **0.807±0.003** |
| CIFAR100 | 0.326±0.008 | 0.399±0.004 | **0.466±0.006** |
| ImageNet1K | 0.539±0.003 | 0.512±0.001 | **0.575±0.004** |

6GELU+6CRReLU is significantly and stably worsen to all GELU and all CRReLU, which is quite surprising to us. We consider that this may be due to the fact that, while the reduction in entropy is not significantly apparent in the earlier layers, CRReLU's focus on achieving lower entropy still facilitates superior feature extraction. It seems that when using GELU in the earlier layers and CRReLU in the later layers, on small-scale datasets, it is still possible to benefit from the CRReLU mechanism in the later layers (the features learned in the earlier layers are not good enough yet); however, on large-scale datasets, the features learned in the earlier layers (those equipped with GELU) might even have a negative effect.

# H  FURTHER DISCUSSION ON INITIALIZATION AND TRAINING STABILITY

## H.1  EXPERIMENTS WITH ADDITIONAL $\varepsilon$ INITIALIZATION

In this section, we focus on exploring the impact of different initial values of $\varepsilon$, as well as potential instabilities or failure cases under different initialization schemes. We set $\varepsilon$ to -0.5, -0.2, -0.1, -0.05, -0.02, -0.01, 0.01, 0.02, 0.05, 0.1, 0.2, 0.5, 1, and 10, conducting experiments with ViT-tiny on CIFAR10 and CIFAR100. We conduct three runs, reporting mean and standard derivation in Table 9.

Table 9: Test accuracy of experiments conducted with ViT for 100 epochs under different initializations.

| $\varepsilon$ | 0.01 | 0.02 | 0.05 | 0.1 | 0.2 | 0.5 | 1 | 10 |
|---|---|---|---|---|---|---|---|---|
| CIFAR10 | 80.7±0.4 | 80.1±0.1 | 79.7±0.3 | 79.6±0.2 | 78.1±0.7 | 74.1±1.0 | 68.7±0.3 | 60.3±0.4 |
| CIFAR100 | 46.6±0.6 | 46.0±0.3 | 45.6±0.4 | 44.9±0.3 | 43.6±0.4 | 36.4±0.9 | 29.9±0.8 | 22.7±0.6 |

| $\varepsilon$ | -0.01 | -0.02 | -0.05 | -0.1 | -0.2 | -0.5 |
|---|---|---|---|---|---|---|
| CIFAR10 | 80.1±0.4 | 80.0±0.3 | 80.1±0.2 | 80.6±0.1 | 80.5±0.3 | 80.4±0.1 |
| CIFAR100 | 45.9±0.3 | 46.0±0.6 | 46.1±0.6 | 46.1±0.3 | 46.0±0.1 | 45.8±0.5 |

From the above experiments, we can see different initialization strategy does can have an impact on the final result. When the initial values differ significantly from the range we derived in Corollary 1 (such as 0.5, 1, 10), we have observed that the performance of training will degrade severely, especially at 1 and 10, where the training process becomes extremely unstable.

## H.2  CHANGE OF $\varepsilon$ DURING TRAINING

In Table 9, we demonstrate that different initialization strategies can have an impact on the final result. Improper initialization can significantly impair performance and lead instability. In this section, we set $\varepsilon$ to 0.01 and show the changes of $\varepsilon$ in each layer during the 100 epochs' training of ViT-tiny on CIFAR10 and CIFAR100 in Figure 4 and Figure 5. As the figures shown, initializing $\varepsilon$ at 0.01 allows its stable change during the training process and could avoid extreme situations.

## H.3  DISCUSSION ON THE INTRODUCED LEARNABLE PARAMETER $\varepsilon$

In Section 4.2, we have successfully demonstrated existence of the worst activation function, and from the worst as a starting point, it always moves towards improvement, regardless of the direction taken. However, commencing from a specific activation function, like ReLU here, does not invariably result in improvement across all directions, i.e. certain optimization paths may lead to deteriorated outcomes. Therefore, from the practical perspective, we introduce learnable parameter $\varepsilon$ with the aim of enabling self-optimization of networks. From another perspective, in the derivation from ReLU

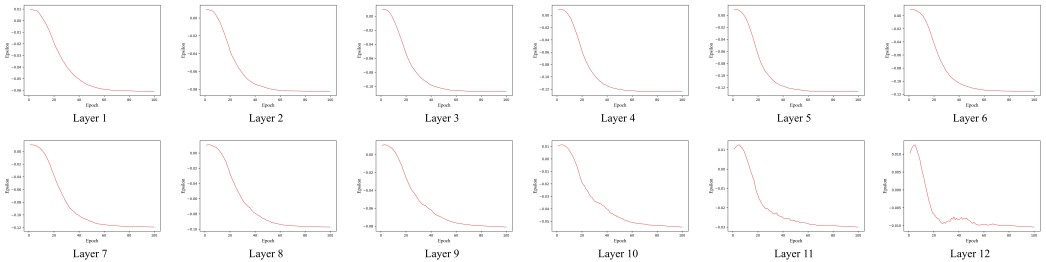

Figure 4: Visualization of $\varepsilon$ change in each layer during the training of ViT-tiny on CIFAR10.

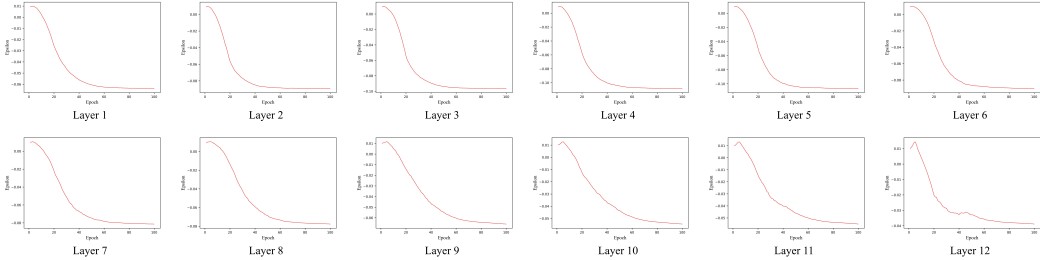

Figure 5: Visualization of $\varepsilon$ change in each layer during the training of ViT-tiny on CIFAR100.

to CRReLU, we assume that data follows Gaussian distribution, which might not be true in real experiments. Existence of the learnable parameter $\varepsilon$ also weakens this assumption to some extent.

Furthermore, we would like to provide some insights on selecting optimal values of $\varepsilon$ for practitioners implementing CRReLU in their own networks. Firstly, we suggest looking within the aforementioned scope (-0.188,0.084) (detailed in Corollary 1). Furthermore, we suggest testing multiple values of it within this range and using k-fold cross-validation to test the performance of model under different initial values. Train and evaluate the model on different folds to find the optimal value. In addition, if the prior knowledge of the dataset and network structure is sufficient, it can also be fully utilized, for example, if the network tends to produce negative outputs, then increasing the value of $\varepsilon$ can be considered to enable the model to better capture the features of the samples.

# I FURTHER DISCUSSION ON LOWER ENTROPY INDICATES BETTER CLASSIFICATION

Our work is largely based on the statement: lower entropy indicates better classification. In this section, we will further elaborate on the reasonability of such a statement. We would like to further elaborate mainly from three perspectives: intuitively, empirically, and theoretically. From the *intuitive* perspective, lower entropy indicates less uncertainty for feature representation, which usually means more information is captured in fewer features. In other words, lower entropy can suggest that features are more discriminative, better able to distinguish different categories or patterns. From the *empirical* perspective, early work (Silva et al., 2005) experimentally showed that minimization of Shannon's entropy of the gap between the output and the desired target could achieve a better performance compared to MSE and CE. In early work (Santos et al., 2004), the authors experimentally illustrated that minimizing entropy of the error between output and desired targets yields exceptionally satisfactory classification performance. From the *theoretical* perspective, work (Yi et al., 2022) proved that for training DNN classifiers essentially learns the conditional entropy of the underlying data distribution of the dataset (the information or uncertainty remained in the labels after revealing the input) and derived the mutual information (between the corresponding feature and the label) bounds for a classification data model (Section 7). Hence, the conditional entropy $\mathbb{H}$(output | input) will decrease with the process of training. In the work (Xu & Raginsky, 2017), the authors derived upper bounds on the generalization error in terms of the mutual information between its input and output. According to (Xu & Raginsky, 2017), a smaller mutual information means a smaller generalization error upper bound, which in turn suggests better classification performance. We have

mutual information $\mathbb{I}(input,output) = \mathbb{H}(output) - \mathbb{H}(output \mid input)$. Within the process of training, $\mathbb{H}(output \mid input)$ decreases; hence, in order to make the mutual information $\mathbb{I}(input,output)$ as small as possible, we should minimize the $\mathbb{H}(output)$. Therefore, we consider that a lower entropy signifies better classification performance.

## J FURTHER DISCUSSION ON BIAS TOWARDS ACTIVATION FUNCTION IN PRE-TRAINED MODELS

### J.1 BIAS EXISTED IN PRE-TRAIN INITIALIZATION

In the experiments of this work, we choose the truncated normal initialization instead of the initialization commonly selected in state-of-the-art (SOTA) methods through pre-training on larger datasets (for instance, ImageNet1K is initialized using weights pre-trained on ImageNet22K). This is mainly based on our following observation. We discover that the pre-trained models that have been released exhibit an intrinsic bias towards the activation functions they utilize (in other words, when the weights of a model pre-trained with GELU are used as initialization, the performance with GELU as the activation function is always optimal. Thus, to facilitate a fair comparison on the activation fuctions, we abandon this initialization method and utilize the truncated normal initialization, which does not introduce any bias on the activation functions.

Furthermore, in the LLM fine-tuning tasks (Table 5), the initial model we utilized is the publicly released GPT-2, which employs the GELU activation function for pre-training. Based on aforementioned observations, the model actually exhibits a bias towards GELU; however, the ultimate results indicate that CRReLU still surpasses GELU, albeit to a lesser extent. Thus, this also demonstrates to some extent the superiority of CRReLU when confronted with larger parameters.

### J.2 BIAS EXISTED IN KNOWLEDGE DISTILLATION

In Table 3, experiments on ViT clearly demonstrate superiority of CRReLU over other activation functions, and those on DieT, GELU shows 0.1% higher mean accuracy compared to CRReLU. Such result is attributed to the teacher-student strategy structure of DieT model. We utilize the fine-tuned "deit-tiny-patch16-224" model as teacher model, which is trained with GELU. As explained in the work (Abnar et al., 2020), through distillation, transformers will inherit inductive bias. Hence, training a student model with GELU on ImageNet-1K with the help of teacher model, which has already been pre-trained on ImageNet-1K with GELU, is certain to achieve better results than other activation functions. Furthermore, it is observed that such issue is not limited to CRReLU actually: when comparing GELU and PReLU, GELU is 2.9% lower than PReLU in the ViT model, whereas in the DeiT model, GELU is 0.9% higher than PReLU. We believe that such issues are related to the bias present in the teacher model towards activation functions (the currently used open-source implementation employs GELU).

## K FURTHER DISCUSSION ON DYNAMIC OPTIMIZATION

As mentioned, the EAFO methodology has shown the potential for dynamic optimization of activation during iterative training. Although we have not yet obtained an effective dynamic optimization method, we still want to provide some discussion here for future research work.

Dynamic optimization during iterative training might introduce considerable computational complexity. Such a problem of computational complexity might require the utilization of more efficient optimization algorithms (or optimizer) to address. Some insights are provided as follows. *Firstly*, we suggest conducting such activation optimization at a "batch-level" (gradient updates are typically done at the mini-batch level), which can stabilize the entire training process on one hand, and on the other hand, can reduce the computational complexity of dynamic optimization. That is to say, we can update the network parameters at a mini-batch level; while updating the activation at a batch level. *Furthermore*, we recommend using techniques similar to momentum methods for the design of optimizers, so that the model can retain information about the speed of gradient descent from the past, thereby accelerating convergence and reducing computing cost overall. *Finally*, we also would like to

consider methods for the adaptive activation learning, similar to Adam, by adjusting the activation learning rates through calculating first and second moment estimates of the gradients.

## L  FURTHER DISCUSSION ON ACTIVATION FUNCTION RANKING

In the past three decades, more than 400 activation functions have been proposed (Kunc & Kléma, 2024). We are thinking about whether we can provide some insights for activation function ranking based on our framework. Here, we would like to provide a little insight through the comparison of information entropy. The information entropy takes the form as:

$$\mathbb{H}(y(x)) = -\int p(y(x))y'(x)\log(p(y(x)y'(x)))dx$$

where $y(x)$ is the inverse function of the activation function.

**Insight 1.** *Under mild assumptions, PReLU with tunable parameters should outperform the Leaky-ReLU with fixed-parameters.*

*Proof of sketch.* For PReLU and Leaky-ReLU, they exhibit similar form: $f(x) = \begin{cases} x & x > 0 \\ \alpha x & x < 0 \end{cases}$. The difference is that $\alpha$ is a tunable parameter for PReLU, whereas a fixed parameter for Leaky-ReLU.

And the inverse functions take the form as: $y(x) = \begin{cases} x & x > 0 \\ x/\alpha & x < 0 \end{cases}$. We further segregate the positive and negative components of entropy function and simplify the expression:

$$\mathbb{H}(y(x)) = -\int_{-\infty}^{0} p(y(x))y'(x)\log(p(y(x)y'(x)))dx - \int_{0}^{+\infty} p(y(x))y'(x)\log(p(y(x)y'(x)))dx$$

$$= -\int_{-\infty}^{0} p(x/\alpha)/\alpha \cdot \log(p(x/\alpha)/\alpha)dx - \int_{0}^{+\infty} p(x)\log(p(x))dx$$

Hence, let's further consider the difference in information entropy between PReLU and Leaky-ReLU:

$$\mathbb{H}(\text{PReLU}) - \mathbb{H}(\text{Leaky-ReLU})$$

$$= -\int_{-\infty}^{0} p(x/\alpha_1)/\alpha_1 \cdot \log(p(x/\alpha_1)/\alpha_1) - p(x/\alpha_2)/\alpha_2 \cdot \log(p(x/\alpha_2)/\alpha_2)dx$$

where $\alpha_1$ represents the tunable parameter of PReLU; $\alpha_2$ represents the fixed parameter of Leaky-ReLU. Moreover, according to the formula, due to the PReLU's ability to dynamically adjust its parameters based on the data distribution $p(\cdot)$, the resulting mutual information will be lower compared to the Leaky-ReLU with fixed parameters, resulting in better classification performance. □

Nevertheless, it is also worthy noting that alteration of parameter $\alpha$ in response to the data distribution will undoubtedly vary across different network architectures. Hence, ranking different activation functions in a generalized condtion is actually a rather challenging task at present stage; for different network architectures, initialization and stochasticity, theoretical understanding on ranking activation functions still requires a considerable amount of discussion and comprehension.

## M  FURTHER DISCUSSION ON LLM INFERENCE TASK

In addition to the LLM fine-tuning tasks, we believe that CRReLU could further potentially achieve a better balance between inference speed and diverse generation in the LLM inference task. In the work (Mirzadeh et al., 2024), it is shown that leveraging the activation sparsity of ReLU, there will be a significant enhancement in inference FLOPS. However, it is also noteworthy that contemporary open-source LLMs increasingly favor the use of GELU and SiLU, likely driven by considerations surrounding the diversity of model generation. Excessive activation sparsity might potentially diminish the generative diversity of the model, thereby reducing user engagement. The authors further illustrate in Figure 2(c) that as the parameter beta increases, the performance of activation sparsity improves. Such observation is closely related to the Lipschitz Continuity of the activation

function (Kim et al., 2021) (last paragraph of Section 3.1 claims that bounded inputs make dot-product self-attention Lipschitz). In Fact 1, we have obtained the GELU's Lipschitz constant of 1.084; Mish's Lipschitz constant of 1.089; SiLU's Lipschitz constant of 1.110. To enhance the performance of CRReLU, resulting in a superior Lipschitz continuity compared to GELU, we derive the recommended range is (-0.188, 0.084). Hence, as CRReLU approaches zero within this range, CRReLU converges more closely to ReLU. In this scenario, activation sparsity of CRReLU improves, while it may also potentially diminish the diversity of the generated outputs. Conversely, as it far away from zero within this range, the utilization of CRReLU could deteriorate activation sparsity, yet simultaneously possess potential to enhance diversity of generated outputs. Therefore, CRReLU has higher flexibility in how much we care about generation diversity compared to activation sparsity.

# N DETAILS OF EXPERIMENTAL SETTINGS

## N.1 TASK OF IMAGE CLASSIFICATION

Table 10: Experimental settings of ViT, DeiT and TNT on CIFAR-10 and CIFAR-100 datasets

| | |
|---|---|
| Image Size | $32 \times 32$ |
| Patch Size | 4 |
| Embedding Dim | 192 for ViT-Tiny and DeiT-Tiny ; 384 for TNT-small |
| Optimizer | AdamW with weight decay = 0.05 |
| Learning Rate | Cosine Annealing Learning Rate Scheduler
Initial lr = $2.5 \times 10^{-4}$ ; lr drop = -1 ; min lr = $1 \times 10^{-5}$ |
| Warm up | warmup epochs = 20 ; warmup learning rate = $1 \times 10^{-6}$ |
| Gradient Clipping | 1.0 |
| Training Epochs | 100 |
| Batch Size | 256 |
| Loss Function | CrossEntropy Loss |
| Normalization | Layer Norm |
| Data Augmentation | True (provided by timm) |
| Drop Out and Drop Path | False |

Table 11: Experimental settings of ViT and DeiT on ImageNet-1K dataset

| | |
|---|---|
| Image Size | $224 \times 224$ |
| Patch Size | 16 |
| Embedding Dim | 192 |
| Optimizer | AdamW with weight decay = 0.05 |
| Learning Rate | Cosine Annealing Learning Rate Scheduler
Initial lr = $2.5 \times 10^{-4}$ ; lr drop = -1 ; min lr = $1 \times 10^{-5}$ |
| Warm up | warmup epochs = 20 ; warmup learning rate = $1 \times 10^{-6}$ |
| Gradient Clipping | 1.0 |
| Training Epochs | 100 |
| Batch Size | 256 |
| Loss Function | CrossEntropy Loss |
| Normalization | Layer Norm |
| Data Augmentation | True (provided by timm) |
| Drop Out and Drop Path | False |

Table 12: We record changes in parameter number when employing various activation functions. GELU, ELU, CELU, SiLU (Swish), and Mish are considered activation functions without learnable parameter **(AFs without LP)**, while PReLU and CRReLU are considered activation functions with learnable parameter **(AFs with LP).** The results demonstrate that increase in parameter number introduced by the learnable parameter is negligible.

| Parameter Number | | CIFAR-10 | CIFAR-100 | ImageNet-1K |
|---|---|---|---|---|
| ViT-Tiny | AFs without LP | 5399818 | 5417188 | 5754472 |
| | AFs with LP | 5399830 | 5417200 | 5754484 |
| DeiT-Tiny | AFs without LP | 5365076 | 5399816 | 5910800 |
| | AFs with LP | 5365088 | 5399828 | 5910812 |
| TNT-Small | AFs without LP | 21525298 | 21559948 | / |
| | AFs with LP | 21525322 | 21559972 | / |

## N.2   TASK OF LARGE LANGUAGE MODEL (LLM) FINE-TUNING

Table 13: Experimental settings of GPT2 fine-tuning task

| | |
|---|---|
| Batch Size | 32 |
| Optimizer | RMSprop (More Memory-Efficient) |
| Learning Rate | $5 \times 10^{-7}$ with linear warmup steps of 150 |
| Trainer | FSDPTrainer (2 GPUs) |
| Max Gradient Norm | 10.0 |
| Max Length for an Input (Prompt + Response) | 512 |
| Max Length for Prompt | 256 |

