# OpenReview forum: "Entropy-based Activation Function Optimization: A Method on Searching Better Activation Functions"
_ICLR.cc/2025/Conference — ICLR 2025 Poster_

### Official Review · Reviewer_m9pv · 2024-10-31

**Soundness:** 3
**Presentation:** 3
**Contribution:** 3
**Rating:** 5
**Confidence:** 4

**Summary:**

The paper presents a systematic approach to address the problem of activation function optimization in artificial neural networks (ANNs). By leveraging information entropy theory, the authors theoretically demonstrate the existence of the worst activation function under boundary conditions (WAFBC). They then propose the Entropy-based Activation Function Optimization (EAFO) methodology, which provides a framework for designing better activation functions. Utilizing this methodology, the authors derive a novel activation function called Correction Regularized ReLU (CRReLU) from the conventional ReLU. Extensive experiments on vision transformer variants and large language model (LLM) fine-tuning tasks demonstrate the superior performance of CRReLU over existing ReLU variants.

**Strengths:**

1. Theoretical Rigor:
The paper provides a solid theoretical foundation for activation function optimization by introducing the concept of WAFBC and the EAFO methodology. This approach is novel and offers a fresh perspective on designing activation functions.
2. Practical Application:
The derived CRReLU activation function shows significant improvements in performance across various tasks, including image classification and LLM fine-tuning, demonstrating the practical applicability of the proposed methodology.
3. Comprehensive Experiments:
The authors conduct extensive experiments on multiple datasets and architectures, validating the effectiveness of CRReLU and providing a thorough evaluation of the proposed method.

**Weaknesses:**

1. Limited Generalizability:
The paper primarily focuses on ReLU and its variants. It would be valuable to explore the applicability of the theoretical framework to activation functions without an inverse function, such as Swish or Mish.
2. Computational Complexity:
The dynamic optimization during iterative training introduces significant computational complexity, which the paper does not address. The authors should discuss potential approaches or algorithms, such as gradient-based optimization or stochastic methods, that might mitigate these computational complexity issues, or provide a more detailed analysis of the trade-offs between performance gains and computational costs.
3. Assumption of Gaussian Distribution:
The assumption that data follows a Gaussian distribution simplifies the derivation of CRReLU but may not hold in all real-world scenarios. The authors should provide empirical evidence or theoretical analysis of CRReLU's performance under non-Gaussian data distributions, such as heavy-tailed or multimodal distributions, to address concerns about the robustness of the method.
4. Lack of Diverse Experiments:
While the experiments are comprehensive, they are limited to specific datasets and architectures. Additional experiments on diverse datasets, such as medical imaging (e.g., MICCAI) or remote sensing data (e.g., EuroSAT), and architectures like convolutional neural networks (e.g., ResNet) or graph neural networks, would strengthen the generalizability claims.

**Questions:**

1. How does the EAFO methodology perform when applied to other activation functions, especially those without an inverse function, such as Swish or Mish?
2. Can the authors provide empirical evidence or theoretical analysis of CRReLU's performance under non-Gaussian data distributions, such as heavy-tailed or multimodal distributions, to address concerns about the robustness of the method?
3. What potential approaches or algorithms, such as gradient-based optimization or stochastic methods, can be explored to mitigate the computational complexity introduced by dynamic optimization during iterative training?
4. Would the authors consider conducting additional experiments on diverse datasets, such as medical imaging (e.g., MICCAI) or remote sensing data (e.g., EuroSAT), and architectures like convolutional neural networks (e.g., ResNet) or graph neural networks, to further validate the generalizability of CRReLU?

**Details Of Ethics Concerns:**

None.

---

> ### Author Response · Authors · 2024-11-21
>
> Dear Reviewer m9pv:
>
> Thank you for your great efforts on our work, for your comprehensive summary on the strengths and weaknesses, for your insightful comments and for your constructive suggestions.
>
> **W1 and Q1**
>
> We wholeheartedly concur with your insightful comments, and thank you once more for them. As stated in your insightful comment, the current EAFO methodology cannot be applied to those without an inverse function. This is discussed as the first point under the "limitations" section, and we intend to leave this part for future work. Our current approach to solving the problem involves adopting the Lebesgue integral form instead of the original Riemann integral utilized in entropy calculation, but more specific implementation and rigorous theoretical derivation are still ongoing.
>
> **W3 and Q2**
>
> Thank you once more for your insightful comments and for your constructive suggestions. In this work, we assume a Gaussian distribution of the data, which has previously been validated in MLPs and CNNs as a reasonable assumption theoretically. However, in more modern architectures, particularly transformers, due to their self-attention mechanisms, this assumption might not hold true. Therefore, we opt to conduct experiments on transformers to ascertain their performance under non-Gaussian distributional conditions. We select to report the experimental outcomes on transformers with the primary aim of providing empirical evidence of CRReLU's performance under non-Gaussian data distributions. We have also considered theoretically analyzing the performance of CRReLU with respect to heavy-tailed distributions or multimodal distributions, but we found that these distributions lack a typical distribution form for further analyze. Furthermore, we consider enhancing our experimental evaluation section. Firstly, we conduct multiple experiments based on the initial experiments in the paper and report the mean and standard deviation to better understand the statistical characteristics of the results (please refer to Response to Reviewer YJtL). Then, following your suggestions, we enhance the evaluation of CRReLU's generalizability to network structures. We have chosen to validate performance of ConvNeXt-tiny [2] (one of the latest CNNs) on CIFAR10, CIFAR100, and ImageNet1K. Experiments on CIFAR10 and CIFAR100 are conducted on 4 RTX3090, those on ImageNet1K are conducted on 4 NVIDIA L20. We perform three runs,  reporting both the mean and standard deviation.
>
> Table1: Test accuracy of experiments conducted on ConvNeXt-tiny for 100 epochs with error bar.
> |            | GELU               | ELU                | PReLU              | CELU               | SiLU               | Mish               | CRReLU             |
> |:----------:|:------------------:|:------------------:|:------------------:|:------------------:|:------------------:|:------------------:|:------------------:|
> | CIFAR10    | 0\.649$\\pm$0\.004 | 0\.598$\\pm$0\.005 | 0\.646$\\pm$0\.014 | 0\.598$\\pm$0\.005 | 0\.606$\\pm$0\.002 | 0\.614$\\pm$0\.004 | **0\.706$\\pm$0\.011** |
> | CIFAR100   | 0\.366$\\pm$0\.003 | 0\.303$\\pm$0\.004 | 0\.352$\\pm$0\.005 | 0\.305$\\pm$0\.002 | 0\.350$\\pm$0\.009 | 0\.353$\\pm$0\.007 | **0\.421$\\pm$0\.007** |
> | ImageNet1K | 0\.729$\\pm$0\.003 | 0\.717$\\pm$0\.005 | 0\.729$\\pm$0\.005 | 0\.718$\\pm$0\.009 | 0\.723$\\pm$0\.007 | 0\.728$\\pm$0\.006 | **0\.732$\\pm$0\.002** |
>
> We hope these additional results could alleviate your concerns to some extent.
>
> **W4 and Q4**
>
> Thank you once more for your insightful comments and for your constructive suggestions. Your constructive suggestions significantly bolster the paper. According to your suggestions, we further conduct additional experiments on diverse datasets and architectures. Firstly, we select to verify the performance of CRReLU on ConvNeXt (one of the latest CNNs), and the results can be found in Global Response. Furthermore, based on your valuable suggestions, we conduct experiments on EuroSAT [1] with ConvNeXt-tiny [2]. All experiments are performed three times, we report the mean and standard deviation. We conduct this part experiments on a single RTX3090 for 25 epochs using the AdamW optimizer, learning rate of 0.0001, cross entropy loss function, batch size of 256. The results are presented in the following table.
>
> Table2: Test accuracy of experiments conducted with ConvNeXt-tiny on the EuroSAT
> |         | GELU              | ELU               | PReLU             | CELU              | SiLU              | Mish              | CRReLU            |
> |:-------:|:-----------------:|:-----------------:|:-----------------:|:-----------------:|:-----------------:|:-----------------:|:-----------------:|
> | EuroSAT | 83\.09$\\pm$1\.06 | 81\.21$\\pm$0\.37 | 81\.33$\\pm$0\.94 | 81\.12$\\pm$0\.27 | 81\.85$\\pm$1\.01 | 82\.23$\\pm$0\.08 | **83\.26$\\pm$0\.52** |

---

> ### Author Response · Authors · 2024-11-21
>
> **W2 and Q3**
>
> Thank you once more for your insightful comments and for your constructive suggestions. We fully agree with your insightful comments, dynamic optimization during iterative training indeed might introduce computational complexity. Such a problem of computational complexity might require the utilization of more efficient optimization algorithms (or optimizer) to address. At present, we have not obtained an algorithm that is sufficiently efficient for large-scale activation optimization. While, we would like to provide some insights. Firstly, we suggest conducting such activation optimization at a "batch-level" (gradient updates are typically done at the mini-batch level), which can stabilize the entire training process on one hand, and on the other hand, can reduce the computational complexity of dynamic optimization. That is to say, we can update the network parameters at a mini-batch level; while updating the activation at a batch level. Furthermore, we recommend using techniques similar to momentum methods for the design of optimizers, so that the model can retain information about the speed of gradient descent from the past, thereby accelerating convergence and reducing computing cost overall. Finally, we also would like to consider methods for the adaptive activation learning, similar to Adam, by adjusting the activation learning rates through calculating first and second moment estimates of the gradients.
>
> Finally, we would like to thank you once again for the your insightful comments, for your constructive suggestions, for your thorough and comprehensive summary on the strengths and weaknesses, and for your great efforts on our work.
>
> Warm regards,
>
> Authors of submission 6110
>
>
> [1] Helber P, Bischke B, Dengel A, et al. Eurosat: A novel dataset and deep learning benchmark for land use and land cover classification[J]. IEEE Journal of Selected Topics in Applied Earth Observations and Remote Sensing, 2019, 12(7): 2217-2226.
>
> [2] Liu Z, Mao H, Wu C Y, et al. A convnet for the 2020s[C]//Proceedings of the IEEE/CVF conference on computer vision and pattern recognition. 2022: 11976-11986.

---

> ### Author Response · Authors · 2024-11-29
> **Sincerely Seeking Your Invaluable Feedback**
>
> Dear Reviewer m9pv:
>
> We hope this message finds you well. As the discussion period draws to a close, we are reaching out to solicit your thoughts on the rebuttal responses and the revised manuscript, inspired by your valuable insights. We have provided additional supportive experiments and conducted further discussions in the rebuttal responses and the revised manuscript.
>
> We would like to briefly summarize the changes we made to the manuscript for your easier navigation. On the additional supportive experiments, specifically, we focus on the following aspects: enhancing all experimental results with three runs, additional architecture (Appendix F.1), additional dataset (Appendix F.2), additional $\epsilon$ initialization (Appendix F.3), entropy calculation after activation (Appendix F.4) and mixed activation function (Appendix F.5). On the additional discussions, we focus on Lipschitz continuity analysis (Appendix G), initialization and training stability (Appendix H), lower entropy indicates better classification (Appendix I) and dynamic optimization (Appendix J).
>
> Your expertise in this domain has been a guiding light in these improvements, and we deeply appreciate your constructive and insightful comments. If there are any remaining questions or concerns, we would be more than happy to discuss further. Could you kindly let us know if the points we addressed resolve your concerns, and if you would consider revisiting your evaluation score based on the additional contents?
>
> Thank you once again for your thoughtful feedback and engagement, as it has greatly contributed to improving the quality of our work.
>
> Warm regards,
>
> Authors of Submission 6110

---

> > ### Author Response · Authors · 2024-12-02
> > **Sincerely Seeking Your Invaluable Feedback**
> >
> > Dear Reviewer m9pv:
> >
> > We hope this message finds you well. As the discussion period draws to a close in 20 hours, we are reaching out to solicit your thoughts on the rebuttal responses and the revised manuscript, inspired by your valuable insights. We have provided additional supportive experiments and conducted further discussions in the rebuttal responses and the revised manuscript.
> >
> > Your feedback is invaluable, and we deeply appreciate your time and effort. If there are any remaining questions or concerns, we would be more than happy to clarify further. Could you kindly let us know if the points we addressed resolve your concerns, and if you would consider revisiting your evaluation score based on the additional evidence?
> >
> > Best regards,
> >
> > Authors of Submission 6110

---

> > > ### Author Response · Authors · 2024-12-03
> > > **Discussion Period draws to a Close in 8 Hours. We are Sincerely Seeking Your Invaluable Feedback.**
> > >
> > > Dear Reviewer m9pv:
> > >
> > > We hope this message finds you well. As the discussion period draws to a close in less than 8 hours, we are reaching out to solicit your thoughts on the rebuttal responses, the revised manuscript and the latest version of the paper (Please download it at the following anonymous link https://anonymous.4open.science/r/Revised_Paper-ICLR2025_6110_submission/ICLR_2025_6110_submission.pdf ). In the latest version, we have:
> > >
> > > 1.	**incorporate all the additional experiments and discussions in the rebuttal**.
> > >
> > > 2.	transition to **a research question-oriented** presentation style, demonstrating a significant improvement in both **content organization** and **presentation clarity**.
> > >
> > > 3.	**balance the length** of the main text and the appendix.
> > >
> > > The following is a more specific elaboration of these modifications.
> > >
> > > In the main text:
> > >
> > > 1.	In the **Introduction**, we present the **three questions** on which the content of this paper is based. And in the **summary part** of Introduction, we show the work we have done to **answer** these three questions.
> > >
> > > 2.	In **Section 4.2**, we give the answer to **Question 1**.
> > >
> > > 3.	In **Section 4.3**, we give the answer to **Question 2**. We change the **presentation form of EAFO methodology outline** in a more aesthetically pleasing manner.
> > >
> > > 4.	In **Section 4.4**, we give the answer to **Question 3**. We add the **main conclusions of Lipschitz Continuity Analysis**.
> > >
> > > 5.	In **Section 5.1**, we add **main results** of **Entropy Analysis across Network Layers**. We **briefly** mentioned **Additional Experiments on Architecture and Dataset**.
> > >
> > > 6.	In the **Discussion**, we provide further discussion on **potential applications**.
> > >
> > > 7.	During writing process, we **cite all content in appendix** in order to facilitate a **good correspondence** for readers.
> > >
> > > In the appendix:
> > >
> > > 1.	We provide detailed **proof of Lipschitz Continuity Analysis** in Appendix E.
> > >
> > > 2.	We provide **additional experiments on architecture and dataset** in Appendix F.
> > >
> > > 3.	We provide **additional experiments on mixed activation function** in Appendix G.
> > >
> > > 4.	We provide **further discussion on initialization and training stability** in Appendix H.
> > >
> > > 5.	We provide **further discussion on lower entropy indicates better classification** in Appendix I.
> > >
> > > 6.	We provide **further discussion on bias towards activation function in pre-trained models** in Appendix J.
> > >
> > > 7.	We provide **further discussion on dynamic optimization** in Appendix K.
> > >
> > > 8.	We provide **further discussion on activation function ranking** in Appendix L.
> > >
> > > 9.	We provide **further discussion on LLM inference task** in Appendix M.
> > >
> > > Your feedback is invaluable, and we deeply appreciate your time and effort. If there are any remaining questions or concerns, we would be more than happy to clarify further. Could you kindly let us know if the points we addressed resolve your concerns, and if you would consider revisiting your evaluation score based on the additional evidence?
> > >
> > > Best regards,
> > >
> > > Authors of Submission 6110

---

### Official Review · Reviewer_LUV6 · 2024-11-03

**Soundness:** 3
**Presentation:** 3
**Contribution:** 2
**Rating:** 8
**Confidence:** 4

**Summary:**

This paper targets the fundamental challenge of activation function design in deep neural networks, which has relied heavily on empirical knowledge rather than a systematic understanding and theoretical foundations. The authors thus propose a new theoretical framework connecting information entropy to activation function performance, which verifies the existence of a worst-case activation function (WAFBC) and thereby develops an entropy-based optimization method (EAFO). The key theoretical contribution of this work is establishing that moving away from WAFBC can consistently improve the model’s performance, leading to a systematic approach for activation function optimization. Built upon this, the authors present Correction Regularized ReLU (CRReLU), demonstrating its great performance across vision transformers and language models. The experiments are comprehensive, covering both image classification (CIFAR-10/100, ImageNet-1K) and language model fine-tuning tasks, with thorough ablation studies and theoretical guarantees.

**Strengths:**

**(S1) Theoretical Foundation:** This paper establishes a solid mathematical framework connecting information entropy to activation function performance. The derivation begins with principles of information theory and extends through functional analysis to establish clear relationships between data distributions and activation behavior. Specifically, the proof of the Worst Activation Function with Boundary Conditions (WAFBC) existence is clear, utilizing variational calculus and the Euler-Lagrange equation to demonstrate global maximality. As such, it not only provides insights into why certain activation functions perform better than others but also explains long-observed empirical phenomena, such as the superior performance of unbounded activation functions (like ReLU) compared to bounded ones (such as sigmoid and tanh), which offers both theoretical guarantees and practical optimization guidance.

**(S2) Technical Originality and Soundness:** The proposed EAFO method represents a significant advancement in activation function design. Unlike previous ones that largely relied on empirical knowledge, EAFO provides a principled and systematic framework. The derivation of correction terms through analysis of the information entropy functional's Taylor expansion is insightful, enabling both static design and potential dynamic optimization. The introduction of learnable parameters in CRReLU demonstrates a thoughtful balance between theoretical purity and practical adaptability. Moreover, its potential extension to dynamic optimization during training seems to open new research directions, while maintaining backward compatibility with existing architectures and optimization techniques.

**(S3) Thorough Experiments:** Experiments in this work are comprehensive and well-designed, covering multiple network architectures and task domains. Extensive ablation studies and sensitivity analyses are also conducted to show the methods’ effectiveness. Concretely, the evaluation across vision transformers (ViT, DeiT, TNT) and LLMs (GPT-2) shows broad applicability, while the performance improvements on classical computer vision benchmarks (like CIFAR-10/100 and ImageNet-1K) provide strong practical validation. The large-scale experiments on language model fine-tuning using Direct Preference Optimization (DPO) provide valuable insights into the method's scalability and generalization capabilities. Moreover, the computational efficiency analysis is particularly useful, showing minimal overhead despite the addition of learnable parameters.

**(S4) Presentation Clarity:**
This manuscript exhibits great clarity in presenting mathematical concepts and empirical results. The progression from theoretical foundations through practical implementation is logical and well-structured, making the work accessible to a broader audience while maintaining technical insights. The mathematical derivations are with appropriate detail and clear step-by-step explanations, facilitating reproducibility and future extensions. In addition, the thorough implementation details, including pseudo-code and network architecture considerations, ensure the practical applicability of this work.

**Weaknesses:**

**(W1) Theoretical Limitations:** The authors make the assumption of Gaussian distribution for the input data distributions in this paper. While it is mathematically convenient, it requires more rigorous justification. While the authors cite the Central Limit Theorem and previous works supporting this assumption in deep neural networks, modern architectures like transformers with complex operators like self-attention mechanisms may exhibit significantly different distribution patterns. This work would benefit from a more detailed analysis of how distribution deviations affect the theoretical guarantees. In addition, the convergence properties during the training process, particularly the interaction between the learnable parameter and standard network weights, lack thorough theoretical treatment.

**(W2) Experimental Concerns:** The experimental results, while generally strong, reveal several areas requiring deeper investigation. The performance compared to GELU in DeiT experiments raises important questions about the interaction between CRReLU and knowledge distillation processes. It deserves a more thorough analysis, potentially exploring alternative distillation strategies that are more compatible with the CRReLU's properties. Besides, the initialization strategy for the learnable parameter appears somewhat arbitrary (set to 0.01). Moreover, the absence of experiments on classical CNN architectures leaves a significant gap in demonstrating the method's generality, particularly given the widespread use of CNN-based network architectures.

**(W3) Dynamic Optimization Challenges:** This work employs dynamic optimization during training, which potentially faces several practical challenges. For example, the computational complexity analysis of dynamic optimization is insufficient, particularly for large-scale networks where activation function optimization could introduce substantial overhead. The interaction between dynamic activation optimization and common training techniques (batch normalization, residual connections, dropout) also requires more detailed analysis. I recommend the authors conduct more experimental validation and analysis to address these issues.

**(W4) Implementation and Scalability Considerations:** The practical implementation of EAFO and CRReLU requires more detailed treatment, particularly regarding numerical stability and computational efficiency at scale. Discussion of potential gradient flow issues when the learnable parameter ε takes extreme values, and the mitigation strategies are all not provided. Additionally, the paper would benefit from analysis of how the method performs under resource-constrained conditions, such as mobile devices or edge computing scenarios. All these could provide more insights to the researchers and practitioners in the community, and thus propel further research.

**Questions:**

**(Q1) Dynamic Optimization Implementation:** While the authors suggest the potential for dynamic optimization of activation functions during training, the practical implementation remains relatively unclear. Could the authors elaborate on:

- Concrete strategies for making dynamic optimization computationally tractable in large networks?
- Specific approaches to balance the frequency of activation function updates with computational overhead?
- Empirical evidence or theoretical bounds on the expected performance gains from dynamic optimization? Understanding these aspects would help assess the practical value of the dynamic optimization extension.

**(Q2) Initialization and Training Stability:** The choice of ε=0.01 as initialization appears somewhat arbitrary. Could the authors provide:

- Analysis of how different initialization values affect training dynamics and final performance?
- Guidelines for selecting optimal ε values based on network architecture or task requirements?
- Can we investigate potential instabilities or failure cases under different initialization schemes? This information would be crucial for practitioners implementing CRReLU in their own networks.

---
**Additional Comment:**

I hope my review helps to further strengthen this paper and helps the authors, fellow reviewers, and Area Chairs understand the basis of my recommendation. I also look forward to the rebuttal feedback and further discussions, and would be glad to raise my rating if thoughtful responses and improvements are provided.


---
## **-------------------- Post-Rebuttal Summary --------------------**

The additional experiments, discussions, and revised manuscript provided by the authors have significantly strengthened the work and addressed most of my concerns. I suppose this work can provide knowledge advancement to the field, and I look forward to the final revised manuscript, incorporating the additional information presented in the rebuttal stage.

---

> ### Author Response · Authors · 2024-11-21
>
> Dear Reviewer LUV6:
>
> Thank you for your great efforts on our work, for your thorough and comprehensive summary on the strengths and weaknesses, for your insightful comments and for your constructive suggestions.
>
> **W1**
>
> We wholeheartedly concur with your insightful comments, and thank you once more for them. In this work, we assume a Gaussian distribution of the data, a hypothesis that has been reasonably validated in previous studies on MLPs and CNNs; thus, our approach for MLPs and CNNs is grounded in theoretical validation. However, as you noted, the self-attention mechanisms in transformers may exhibit significantly different distribution patterns, and no studies have yet elucidated the precise form of these distributions. Consequently, we employ transformers in our experiments, relying on empirical validation. We believe that current research on the distribution patterns of modern architectures is still limited and insufficient to support the theoretical guarantees regarding how distribution deviations impact this work. Therefore, we intend to leave this issue for future exploration.
>
> Regarding the convergence properties, we would like to offer some additional insight. We would like to further analyze the Lipschitz continuity of them. In prior work [1][2][3][4], it is shown that Lipschitz continuity exerts a significant influence on convergence, and in work [5], the authors demonstrate the Lipschitz continuity of GELU and computes its Lipschitz constant.
>
> **Defination** A function $f(x)$ is said to be Lipschitz continuous if there exists a constant $L\\geq0$ such that for all x, y $\\in$ R, the following inequality holds:
>
> \\[
> |f(x)-f(y)| \\leq L |x-y|
> \\]
>
> Moreover, a smaller Lipschitz constant indicates a higher degree of Lipschitz continuity.
>
> In work [5], the authors compute Lipschitz constant by finding absolute value of the derivative of GELU function. And the Lipschitz constant is computed to be 1.084. Furthermore, we intend to initially compute the Lipschitz constants of SiLU and Mish.
>
> **Insight1** Lipschitz constant of SiLU is 1.09984.
>
> proof of scratch:
>
> \\[SiLU(x)=\\frac{x}{1+e^{-x}}\\]
> \\[
> \\frac{d SiLU(x)}{d x}=\\frac{(x+1)e^{-x}+1}{(1+e^{-x})^2}
> \\]
> \\[
> \\frac{d^{2}SiLU(x)}{d x^2}=\\frac{e^x(-e^x(x-2)+x+2)}{(1+e^{-x})^3}
> \\]
> Upon further calculations, we ascertain that $\\frac{d SiLU(x)}{d x}$ is bounded in the range of [-0.0998,1.0998].
>
> **Insight2** Lipschitz constant of Mish is 1.0885.
>
> proof of scratch:
>
> \\[Mish(x)=x\\frac{e^{2x}+2e^x}{e^{2x}+2e^x+2}\\]
> \\[
> \\frac{d Mish(x)}{d x}=\\frac{e^x[4(x+1)+4e^{2x}+e^{3x}+e^x(4x+6)]}{(e^{2x}+2e^x+2)^2}
> \\]
> \\[
> \\frac{d^{2} Mish(x)}{d x^2}=\\frac{4e^x(3e^{2x}(x-2)+2e^{3x}(x-1)-2(x+2)-2e^x(x+4))}{(e^{2x}+2e^x+2)^3}
> \\]
> Upon further calculations, we ascertain that $\\frac{d Mish(x)}{d x}$ is bounded in the range of [-0.112526,1.0885].
>
> **Insight3** Lipschitz constant of CRReLU is $\\max(1+\\epsilon, 1-0.446\\epsilon)$.
>
> proof of scratch:
>
> CRReLU$(x)=x+\\epsilon x e^{-\\frac{x^2}{x}} (x \\succ 0)$ and $\\epsilon x e^{-\\frac{x^2}{x}}(x \\prec 0)$.
> Under mild assumptions, we consider the derivative of CRReLU piecewise.
>
> \\[
> \\frac{d CRReLU(x)}{d x}=1+\\epsilon(1-x^2)e^{-\\frac{x^2}{2}} (x \\succ 0) and \\frac{d CRReLU(x)}{d x}= \\epsilon(1-x^2)e^{-\\frac{x^2}{2}} (x \\prec 0)
> \\]
> Setting its second derivative to be 0 (temporary disregarding the potential for non-differentiability at x = 0):
> \\[
> \\frac{d^2 CRReLU(x)}{d x^2} = \\epsilon e^{-\\frac{x^2}{2}} (x^3-3x)=0
> \\]
> Then we have: $x_1=0, x_2=\\sqrt{3}, x_3=-\\sqrt{3}$.
>
>
> When taking $x_2$ in, $\\frac{d CRReLU(x)}{d x}=1-0.446\\epsilon$; when taking $x_3$ in, $\\frac{d CRReLU(x)}{d x}=-0.446\\epsilon$
>
>
> Considering the need to ascertain upper bound for its derivative, we will take into account both sides' values at
> $x=0$. Hence, under mild assumptions,
> \\[
> L=\\max(1+\\epsilon, \\epsilon, 1-0.446\\epsilon, -0.446\\epsilon)
> \\]
> Further consider it, if $\\epsilon\\succ 0$, we have $L=1+\\epsilon$; and if $\\epsilon \\prec 0$, we have $L=1-0.446\\epsilon$. Hence, we can express Lipschitz constant of CRReLU as $\max(1+\\epsilon, 1-0.446\\epsilon)$.
>
> **Insight4** In order to make Lipschitz constant of CRReLU remains lower than that of GELU, the range of $\\epsilon$ is [-0.188,0.084]. We recommend setting the initial value of $\\epsilon$ within this range.

---

> ### Author Response · Authors · 2024-11-21
>
> **W2**
>
> Thank you once more for your insightful comment and for your constructive suggestions. We further conduct multiple runs of all experiments, reporting the error bars to elucidate statistical properties of the results. Furthermore, we conduct experiments on ConvNeXT, the latest CNN architecture on EuroSAT, CIFAR10, CIFAR100 and ImageNet1K. Please refer to the results in Response to Reviewer m9pv. In response to Reviewer YJtL, we provide the entropy calculations across all 12 layers of ViT and the experimental results with 6 layers using GELU and 6 layers using CRReLU. We hope these additional experimental enhancements could alleviate your concerns to some extent. The point you mentioned about the interaction between CRReLU and knowledge distillation processes is a quite insightful comment. Furthermore, we can observe that such issues are not limited to CRReLU: when comparing GELU and PReLU, GELU is 3\% lower than PReLU in the ViT model, whereas in the DeiT model, GELU is 0.7\% higher than PReLU. We believe that such issues are more related to the bias present in the teacher model towards activation functions (the currently used open-source implementation employs GELU), so the point you mentioned is applicable to all activation functions except GELU. In other words, in this context, this is not a fair comparison when other activation functions are being compared to GELU. Regarding the initialization strategy, we would like to further present insights in response to Q2.
>
> **Q2**
>
> Thank you once more for your insightful comments and for your constructive suggestions. As previously mentioned, we believe that it is a better choice for $\epsilon$ within the scope of [-0.188,0.084], which would make the CRReLU's Lipschitz continuity better than GELU's. Furthermore, with computation, if $\epsilon$ is in [0.084,0.0885] and [-0.198, -0.188], we have the CRReLU's Lipschitz continuity worsen than GELU, but better than Mish. And if $\epsilon$ is in [0.0885,0.0998] and [-0.2238,-0.198], we have the CRReLU's Lipschitz continuity worsen than Mish, but better than SiLU.  Following your suggestions, we further conduct experiments with vit-tiny on CIFAR10 and CIFAR100, setting $\\epsilon$ to different initial values: -0.5, -0.2, -0.1, -0.05, -0.02, -0.01, 0.01, 0.02, 0.05, 0.1, 0.2, 0.5, 1, and 10. We conduct three runs under each condition and report mean and standard deviation.
>
> Table1: Test accuracy of experiments conducted with ViT for 100 epochs under different initializations with error bar
> | $\\epsilon$ | 0\.01              | 0\.02              | 0\.05              | 0\.1               | 0\.2               | 0\.5               | 1                  | 10                 |
> |:-----------:|:------------------:|:------------------:|:------------------:|:------------------:|:------------------:|:------------------:|:------------------:|:------------------:|
> | CIFAR10     | 0\.807$\\pm$0\.004 | 0\.801$\\pm$0\.001 | 0\.797$\\pm$0\.003 | 0\.796$\\pm$0\.002 | 0\.781$\\pm$0\.007 | 0\.741$\\pm$0\.010 | 0\.687$\\pm$0\.003 | 0\.603$\\pm$0\.004 |
> | CIFAR100    | 0\.466$\\pm$0\.006 | 0\.460$\\pm$0\.003 | 0\.456$\\pm$0\.004 | 0\.449$\\pm$0\.003 | 0\.436$\\pm$0\.004 | 0\.364$\\pm$0\.009 | 0\.299$\\pm$0\.008 | 0\.227$\\pm$0\.006 |
>
>
> | $\\epsilon$ | \-0\.01            | \-0\.02            | \-0\.05            | \-0\.1             | \-0\.2             | \-0\.5             |
> |:-----------:|:------------------:|:------------------:|:------------------:|:------------------:|:------------------:|:------------------:|
> | CIFAR10     | 0\.801$\\pm$0\.004 | 0\.800$\\pm$0\.003 | 0\.801$\\pm$0\.002 | 0\.806$\\pm$0\.001 | 0\.805$\\pm$0\.003 | 0\.804$\\pm$0\.001 |
> | CIFAR100    | 0\.459$\\pm$0\.003 | 0\.460$\\pm$0\.006 | 0\.461$\\pm$0\.006 | 0\.461$\\pm$0\.003 | 0\.460$\\pm$0\.001 | 0\.458$\\pm$0\.005 |
>
> From the above experiment, we can see different initialization strategy does can have an impact on the final result. When the initial values differ significantly from the values we derived earlier (such as 0.5, 1, 10), we have observed that the performance of training will degrade severely, especially at 1 and 10, where the training process becomes extremely unstable. On the guidelines for selecting optimal values of $\epsilon$, we would like to provide some insights. Firstly, we suggest looking within the aforementioned scope [-0.188,0.084]. Furthermore, we suggest testing multiple values of it within this range and using k-fold cross-validation to test the performance of model under different initial values. Train and evaluate the model on different folds to find the optimal value. In addition, if the prior knowledge of the dataset and network structure is sufficient, it can also be fully utilized, for example, if the network tends to produce negative outputs, then increasing the value of $\epsilon$ can be considered to enable the model to better capture the features of the samples.

---

> ### Author Response · Authors · 2024-11-21
>
> **W3 and Q1**
>
> Thank you once more for your insightful comments and for your constructive suggestions. We fully agree with the point you raised that extensively using activation dynamics optimization in large-scale neural networks could likely result in enormous computational costs. We discuss this point under the second limitation, and it is still an issue that we are actively researching. In this paper, we resort to optimization with learnable parameters for such dynamic optimization, but as of now, we do not have an algorithm that can effectively perform dynamic optimization of activation, and it seems that no such algorithm has been developed within the community either. We intend to leave this challenging problem for future work. In addition, we are considering designing algorithms under network structures that inherently focus on the optimization of activation functions, such as KANs. Regarding the second point you mentioned, we plan to set the frequency of activation function updates at the batch-level, which not only helps to optimize the reduction in execution computation but also increases the stability of training.
>
> **W4**
>
> Thank you once more for your insightful comments and for your constructive suggestions. Regarding the issue of numerical stability, as shown in the aforementioned table, when $\\epsilon$ takes on an extreme value (such as initializing to 10), there is a dramatic decrease in performance and instability in training, therefore, your viewpoint is completely correct. Furthermore, we believe that an appropriate initialization can mitigate this issue: by initializing it within the recommended range, we observe that the change of $\\epsilon$ during the entire training process (initialized as 0.01) remains between -0.2 and 0.02.
>
> Finally, we would like to thank you once again for the your insightful comments, for your constructive suggestions, for your thorough and comprehensive summary on the strengths and weaknesses, and for your great efforts on our work.
>
> Warm regards,
>
> Authors of submission 6110
>
>
> [1] Gouk H, Frank E, Pfahringer B, et al. Regularisation of neural networks by enforcing lipschitz continuity[J]. Machine Learning, 2021, 110: 393-416.
>
> [2] Khromov G, Singh S P. Some Fundamental Aspects about Lipschitz Continuity of Neural Networks[C]//The Twelfth International Conference on Learning Representations. 2024.
>
> [3] Xu Y, Zhang H. Uniform Convergence of Deep Neural Networks with Lipschitz Continuous Activation Functions and Variable Widths[J]. IEEE Transactions on Information Theory, 2024.
>
> [4]Béthune L. Deep learning with Lipschitz constraints[D]. Université de Toulouse, 2024.
>
> [5]Lee M. Gelu activation function in deep learning: a comprehensive mathematical analysis and performance[J]. arXiv preprint arXiv:2305.12073, 2023.
>
> [6] Liu Z, Mao H, Wu C Y, et al. A convnet for the 2020s[C]//Proceedings of the IEEE/CVF conference on computer vision and pattern recognition. 2022: 11976-11986.

---

> ### Author Response · Authors · 2024-11-29
> **Sincerely Seeking Your Invaluable Feedback**
>
> Dear Reviewer LUV6:
>
> We hope this message finds you well. As the discussion period draws to a close, we are reaching out to solicit your thoughts on the rebuttal responses and the revised manuscript, inspired by your valuable insights. We have provided additional supportive experiments and conducted further discussions in the rebuttal responses and the revised manuscript.
>
> We would like to briefly summarize the changes we made to the manuscript for your easier navigation. On the additional supportive experiments, specifically, we focus on the following aspects: enhancing all experimental results with three runs, additional architecture (Appendix F.1), additional dataset (Appendix F.2), additional $\epsilon$ initialization (Appendix F.3), entropy calculation after activation (Appendix F.4) and mixed activation function (Appendix F.5). On the additional discussions, we focus on Lipschitz continuity analysis (Appendix G), initialization and training stability (Appendix H), lower entropy indicates better classification (Appendix I) and dynamic optimization (Appendix J).
>
> Your expertise in this domain has been a guiding light in these improvements, and we deeply appreciate your constructive and insightful comments. If there are any remaining questions or concerns, we would be more than happy to discuss further. Could you kindly let us know if the points we addressed resolve your concerns, and if you would consider revisiting your evaluation score based on the additional contents?
>
> Thank you once again for your thoughtful feedback and engagement, as it has greatly contributed to improving the quality of our work.
>
> Warm regards,
>
> Authors of Submission 6110

---

> > ### Comment · Reviewer_LUV6 · 2024-11-30
> > **Official Response from Reviewer LUV6 to the Rebuttal**
> >
> > Dear Authors of Submission 6110,
> >
> > I have thoroughly reviewed the authors' responses and carefully examined all the additional experimental results provided in the rebuttal. After careful consideration of both the rebuttal and the revised manuscript, I find that the authors have made substantial improvements that address the key weaknesses identified in my original comments.
> >
> > The theoretical foundation has been significantly strengthened through the addition of rigorous Lipschitz continuity analysis. The authors have meticulously derived and compared the Lipschitz constants for multiple activation functions, including GELU (1.084), Mish (1.089), SiLU (1.09984), and CRReLU (max(1+ε, 1-0.446ε)). This analysis led to a well-justified recommendation for the initialization range of ε ∈ [-0.188, 0.084], providing crucial practical guidance for implementation.
> >
> > The authors have also provided thoughtful responses regarding the dynamic optimization challenges, suggesting practical approaches like batch-level updates and momentum-based techniques. While full dynamic optimization remains an open challenge, the proposed strategies offer viable paths forward.
> >
> > Regarding the Gaussian distribution assumption, the authors acknowledge its limitations while providing empirical evidence of CRReLU's effectiveness even in scenarios where this assumption may not hold, particularly in Transformer-based network architectures.
> >
> > Given these substantial improvements and clarifications, I am revising my rating from 5 to 6, as the work now presents a more complete contribution to the field. I strongly recommend the author **incorporate all the additional experiments and discussions in the rebuttal to the revised manuscript** to enhance its soundness. I look forward to further discussions with the authors.
> >
> > Best regards,
> >
> > Reviewer LUV6

---

> > > ### Author Response · Authors · 2024-11-30
> > > **Thank you for the Invaluable Feedback and Precious Suggestions**
> > >
> > > Dear Reviewer LUV6:
> > >
> > > We would like to thank you once more for your great efforts and time on our work, for your thorough and comprehensive summary on the strengths and weaknesses, for your insightful comments and for your constructive suggestions. We are carefully working on summarizing all the additional experiments and discussions in the rebuttal and will incorporate all of them in a more coherent and logical framework in the next version. Concurrently, we will carefully consider the balance between the primary text and the appendix to ensure that the work remains comprehensible to a wider audience without compromising on technical profundity.
> > >
> > > We will incorporate all of your precious suggestions into the next version of the paper; they are indeed profoundly beneficial. Your expertise in this domain is highly admirable, and we also look forward to further discussions with you.
> > >
> > > Finally, we would like to thank you once again for your invaluable feedback and precious suggestions on the manuscript.
> > >
> > > Best regards,
> > >
> > > Authors of Submission 6110

---

> > > > ### Comment · Reviewer_LUV6 · 2024-11-30
> > > > **Official Response by Reviewer LUV6 to the Authors**
> > > >
> > > > Dear Authors,
> > > >
> > > > Thank you for your kind acknowledgment. I appreciate that my suggestions can help strengthen this work further. In particular, I am pleased with your commitment to balancing technical depth with broad accessibility - this is crucial for maximizing the paper's impact on the field.
> > > >
> > > > Built upon this, from my perspective, I would further recommend reframing the manuscript to more explicitly emphasize the fundamental challenges that CRReLU addresses, particularly regarding the optimization and design of activation functions in deep neural networks. Transitioning the paper's presentation style from a largely method-oriented way to a **research question-oriented** narrative would better highlight the significant contributions of this entropy-based activation framework to the broader research community. This reframing would more effectively communicate the **key advances in knowledge** of this work, which ultimately provides enduring value for both researchers and practitioners in the community.
> > > >
> > > > I look forward to seeing the next version of the revised manuscript and am confident that these additions will significantly enhance its value, whether for the current phase or future submissions. I remain actively engaged in this review process and encourage you to reach out if you need any clarification regarding my previous suggestions.
> > > >
> > > > Best regards,
> > > >
> > > > Reviewer LUV6

---

> ### Author Response · Authors · 2024-12-02
> **Thank you for the Precious Suggestions.  We have completed Next Version of the Paper.**
>
> Dear Reviewer LUV6:
>
> We would like to thank you once more for your invaluable feedback and precious suggestions on the manuscript. Based on your suggestions, we have completed the next version of the paper. Please download the latest version of the paper at https://anonymous.4open.science/r/Revised_Paper-ICLR2025_6110_submission/ICLR_2025_6110_submission.pdf. In this version, we make some changes to the paper's presentation style and adjust the text colors to some extent, so we do not use color to indicate modified content with the purpose of avoiding color confusion. In this version of paper, we:
>
> 1.	incorporate all the additional experiments and discussions in the rebuttal
>
> 2.	transition presentation style from a largely method-oriented way to a research question-oriented
>
> 3.	balance the length of the main text and the appendix
>
> The following is a more specific elaboration of these modifications.
>
> In the main text:
>
> 1.	In the Introduction, we present the three questions on which the content of this paper is based. And in the summary part of Introduction, we show the work we have done to answer these three questions.
> 2.	In Section 4.2, we give the answer to Question 1.
> 3.	In Section 4.3, we give the answer to Question 2. We change the presentation form of EAFO methodology outline in a more aesthetically pleasing manner.
> 4.	In Section 4.4, we give the answer to Question 3. We add the main conclusions of Lipschitz Continuity Analysis.
> 5.	In Section 5.1, we add main results of Entropy Analysis across Network Layers. We briefly mentioned Additional Experiments on Architecture and Dataset.
> 6.	In the Discussion, we provide further discussion on potential applications.
> 7.	During writing process, we cite all content in appendix in order to facilitate a good correspondence for readers.
>
> In the appendix:
>
> 1.	We provide detailed proof of Lipschitz Continuity Analysis in Appendix E.
>
> 2.	We provide additional experiments on architecture and dataset in Appendix F.
>
> 3.	We provide additional experiments on mixed activation function in Appendix G.
>
> 4.	We provide further discussion on initialization and training stability in Appendix H.
>
> 5.	We provide further discussion on lower entropy indicates better classification in Appendix I.
>
> 6.	We provide further discussion on bias towards activation function in pre-trained models in Appendix J.
>
> 7.	We provide further discussion on dynamic optimization in Appendix K.
>
> 8.	We provide further discussion on activation function ranking in Appendix L.
>
> 9.	We provide further discussion on LLM inference task in Appendix M.
>
> In the next version, we will continue to strengthen the connections between sections and polish the language of our paper.
>
> Finally, we would like to thank you once again for your invaluable feedback and precious suggestions on the manuscript.
>
> Best regards,
>
> Authors of Submission 6110

---

> > ### Comment · Reviewer_LUV6 · 2024-12-03
> > **Official Response by Reviewer LUV6 to the Authors**
> >
> > Dear Authors,
> >
> > I have thoroughly reviewed your latest response and the revised manuscript with the changes you've outlined. I am pleased to see the comprehensiveness of the revisions, particularly the transition to a research question-oriented presentation style, which demonstrates a significant improvement in both **content organization** and **presentation clarity**. The enhanced logical flow effectively conveys the key research questions and contributions of this work to the field.
> >
> > The restructuring of the paper around fundamental questions in the Introduction, with corresponding answers developed through Sections 4.2-4.4, significantly enhances the paper's logical flow and accessibility. The addition of **Lipschitz Continuity Analysis** in Section 4.4 and the **Entropy Analysis across Network Layers** in Section 5.1 addresses key concerns I had raised previously, providing robust foundations for the methodology.
> >
> > I am particularly impressed with the revised appendix, which now provide **detailed supporting evidence** while maintaining excellent readability in the main text. The additional discussions on initialization, training stability, and practical applications demonstrate both theoretical rigor and practical applicability of the proposed method.
> >
> > In light of these improvements, I am revising my rating from 6 to 8, as I suppose this may better reflect the current overall quality of this paper. I remain available for further discussions.
> >
> > Best regards,
> >
> > Reviewer LUV6

---

### Official Review · Reviewer_7aze · 2024-11-03

**Soundness:** 3
**Presentation:** 3
**Contribution:** 3
**Rating:** 6
**Confidence:** 3

**Summary:**

The paper introduces a theoretical framework to learn a high-performance activation function. It theoretically shows that a worst activation function exist and empirically show that their proposed framework learns significantly improved activation functions compared to SoTA activation function.

**Strengths:**

The proposed framework to learn activation is shown to perform significantly better than the SoTA activation function theoretically as well as empirically.

While different networks and tasks require different activation functions for the best performance, the proposed framework simplifies this design choice by transferring the activation choice to automated learning during the optimization stage.

**Weaknesses:**

Although the paper demonstrates substantial empirical improvements, the reported results fall considerably short of state-of-the-art (SoTA) baseline accuracies. For instance, CNNs using ReLU activation commonly achieve test scores above 0.9.

Additionally, there is no direct comparison between SoTA neural network architectures (such as ViT and CNN) using their standard activation functions and those with the proposed activation function. This makes it unclear how much the new activation function improves upon SoTA.

In the LLM fine-tuning task, the improvement over GeLU activation is minimal.

**Questions:**

Why is the baseline accuracy on ImageNet and CIFAR-10 so low? State-of-the-art networks typically achieve test scores over 0.9 on CIFAR-10 and above 0.8 on ImageNet-1K.

In LLM fine-tuning tasks, the paper reports marginal improvements over GELU. Could the authors provide further insight into the specific benefits of CRReLU in this context, beyond numerical accuracy improvements?

How would CRReLU perform if evaluated on more diverse NLP tasks or models with larger parameters, and would any tuning adjustments be needed?

---

> ### Author Response · Authors · 2024-11-21
>
> Dear Reviewer 7aze:
>
> Thank you for your efforts on our work, for your insightful comments and for your constructive suggestions. Based on your comments, we summarize the strength of our work as follows:
> 1. It performs **significantly better** than SOTAs.
> 2. It exhibits **good generalization** towards different networks and tasks.
>
> And the weaknesses as follows:
>
> 1. The reported results fall considerably short of state-of-the-art (SoTA) baseline accuracies.
> 2. For LLM fine-tuning task, the improvement over GELU activation is minimal.
>
> **W1 and Q1**
>
> Thank you once more for your insightful comment and for your constructive suggestions.  The primary reason for the low baseline accuracy on ImageNet and CIFAR-10 lies in the initialization method employed. In the results, we employ the 'trunc-normal' initialization for (such as line216\~221,line263\~278 in the code".\EAFO-code\EAFO-Image\_classification\reconstruction\models\vit.py").
>
> In reporting SOTA achievements, researchers often employ initialization through pre-training on larger datasets (for instance, ImageNet1K is initialized using weights pre-trained on ImageNet22K). In the process of executing such initialization, we discover that the pre-trained models they have released exhibit an intrinsic bias towards the activation functions they utilize (In other words, a model pre-trained with GELU seems consistently approach better performance of GELU in understream tasks).  Thus, to facilitate a fair comparison on the activation fuctions, we abandon this initialization method and utilize the 'trunc-normal' initialization, which does not introduce any bias on the activation functions. If we are to further compare these activation functions with the reported state-of-the-art results, it is essential to pre-train each activation function individually, which incurs prohibitively high costs. Our work primarily focuses on comparing the empirical performances of different activation functions; thus, we opted to forego the pre-training initialization method. Furthermore, we enhanced the experimental results presented in the paper by conducting multiple experiments (please refer to Response to Reviewer YJtL) , report the results of ConvNeXt on CIFAR and ImageNet1K and the results on the EuroSAT dataset with ConvNeXt (please refer to Response to Reviewer m9pv).  We hope these additional experimental results can help mitigate your concerns.
>
> **W2, Q2 and Q3**
> Thank you once more for your insightful comment and for your constructive suggestions. In the LLM fine-tuning tasks, the initial model we utilized is the publicly released GPT-2, which employs the GELU activation function for pre-training. Based on our previous observations, the model exhibits a bias towards GELU; however, the ultimate results indicate that CRReLU still surpasses GELU, albeit to a lesser extent. Thus, this also demonstrates to some extent the superiority of CRReLU when confronted with larger parameters.
>
> Furthermore, CRReLU could potentially achieve a better balance between inference speed responses and diverse generation in LLM inference tasks. In the work [1], the authors show that leveraging the activation sparsity of ReLU, there will be a significant enhancement in inference FLOPS. However, it is also noteworthy that contemporary open-source LLMs increasingly favor the use of GELU and SiLU, likely driven by considerations surrounding the diversity of model generation. Excessive activation sparsity might potentially diminish the generative diversity of the model, thereby reducing user engagement. The authors further illustrate in Figure 2(c) that as the parameter beta increases, the performance of activation sparsity improves. Such observation is closely related to the Lipschitz Continuity of activation function [2](last paragraph of Section 3.1 claims that bounded inputs make dot-product self-attention Lipschitz).  In response to Reviewer LUV6, we show a detailed examination of the Lipschitz continuity of GELU, SiLU, Mish and CRReLU. In summary, we have obtained the GELU's Lipschitz constant of 1.084; Mish's Lipschitz constant of 1.089; SiLU's Lipschitz constant of 1.09984. To enhance the performance of CRReLU, resulting in a superior Lipschitz continuity compared to GELU, we derive that $-0.188 \leq  \epsilon \leq 0.084$. We recommend that when applying CRReLU, the initialization parameters be set within this range. As it approaches zero within this range, CRReLU converges more closely to ReLU. According to [2], activation sparsity improves, while it may also potentially diminish the diversity of the generated outputs. Conversely, as it far away from  zero within this range, the utilization of CRReLU may deteriorate activation sparsity, yet simultaneously possess potential to enhance diversity of generated outputs.
>
> Finally, we would like to thank you once again for the constructive suggestion and insightful comments on our work.
>
> Warm regards,
>
> Authors of submission 6110

---

> ### Author Response · Authors · 2024-11-21
>
> [1] Mirzadeh S I, Alizadeh-Vahid K, Mehta S, et al. ReLU Strikes Back: Exploiting Activation Sparsity in Large Language Models[C]//The Twelfth International Conference on Learning Representations.
>
> [2] Kim H, Papamakarios G, Mnih A. The lipschitz constant of self-attention[C]//International Conference on Machine Learning. PMLR, 2021: 5562-5571.

---

> ### Author Response · Authors · 2024-11-29
> **Sincerely Seeking Your Invaluable Feedback**
>
> Dear Reviewer 7aze:
>
> We hope this message finds you well. As the discussion period draws to a close, we are reaching out to solicit your thoughts on the rebuttal responses and the revised manuscript, inspired by your valuable insights. We have provided additional supportive experiments and conducted further discussions in the rebuttal responses and the revised manuscript.
>
> We would like to briefly summarize the changes we made to the manuscript for your easier navigation. On the additional supportive experiments, specifically, we focus on the following aspects: enhancing all experimental results with three runs, additional architecture (Appendix F.1), additional dataset (Appendix F.2), additional $\epsilon$ initialization (Appendix F.3), entropy calculation after activation (Appendix F.4) and mixed activation function (Appendix F.5). On the additional discussions, we focus on Lipschitz continuity analysis (Appendix G), initialization and training stability (Appendix H), lower entropy indicates better classification (Appendix I) and dynamic optimization (Appendix J).
>
> Your expertise in this domain has been a guiding light in these improvements, and we deeply appreciate your constructive and insightful comments. If there are any remaining questions or concerns, we would be more than happy to discuss further. Could you kindly let us know if the points we addressed resolve your concerns, and if you would consider revisiting your evaluation score based on the additional contents?
>
> Thank you once again for your thoughtful feedback and engagement, as it has greatly contributed to improving the quality of our work.
>
> Warm regards,
>
> Authors of Submission 6110

---

> > ### Author Response · Authors · 2024-12-02
> > **Sincerely Seeking Your Invaluable Feedback**
> >
> > Dear Reviewer 7aze:
> >
> > We hope this message finds you well. As the discussion period draws to a close in 20 hours, we are reaching out to solicit your thoughts on the rebuttal responses and the revised manuscript, inspired by your valuable insights. We have provided additional supportive experiments and conducted further discussions in the rebuttal responses and the revised manuscript.
> >
> > Your feedback is invaluable, and we deeply appreciate your time and effort. If there are any remaining questions or concerns, we would be more than happy to clarify further. Could you kindly let us know if the points we addressed resolve your concerns, and if you would consider revisiting your evaluation score based on the additional evidence?
> >
> > Best regards,
> >
> > Authors of Submission 6110

---

> > > ### Comment · Reviewer_7aze · 2024-12-02
> > >
> > > Dear Authors
> > >
> > > Thanks for your detailed answers. I am satisfied with the answers and am willing to increase my score as well.

---

### Official Review · Reviewer_YJtL · 2024-11-04

**Soundness:** 2
**Presentation:** 3
**Contribution:** 2
**Rating:** 5
**Confidence:** 3

**Summary:**

In this work, the authors propose a theoretical framework for defining optimality of an activation function (without the optimization considerations). Using Taylor's expansion, the authors extend their framework to search for better activation functions (EAFO - Entropy based Activation Function Optimization) and later also define the worst activation function with boundary conditions. Using the EAFO framework and starting from ReLU, the authors derive a better and novel activation function CRReLU (Correction Regularized ReLU). The authors later demonstrate on three datasets CIFAR10, CIFAR100 and ImageNet-1k where the new found activation function outperforms ReLU on classification performance. Lastly, authors also show improved performance on LLM fine tuning tasks when the CRReLU was swapped out with the ReLU activation function.

**Strengths:**

1. The paper is written clearly and concisely, and is easy to read.
2. An information theoretic framework for defining optimality of activation functions for classification tasks is a great approach to search for activation functions and could potentially generate insights. The authors indicate several properties of worst activation functions in i.e. being bounded however this might require more careful analysis but serves as a good stepping stone for future follow up works.

**Weaknesses:**

1. The premise for EAFO is that extremas in the entropy space after transformation with the activation function correspond to better separability of features in the resulting space but that doesn’t mean better classification performance. Moreover, unlike in discrete space the entropy in the continuous random variables also changes with the scale. However, that might not have any impact on the classification performance. Why do the authors believe this is the right measure to define how good an activation function is?
2. Can the authors rank different activation functions based on the EAFO framework? For example, comparison of ReLU and PReLU should point to PReU being better. Since there is already experimental evidence that PReLU is better, if EAFO could confirm it, that would be a great contribution. Similarly please consider ranking 3-4 activation functions to justify the utility of this framework.
3. For the experiments, what are the error bars? How many training runs per result? This is important to understand the statistical significance of the results.

**Questions:**

1. My main concern regarding the manuscript is—entropy as an indicator of better classification seems like a very strong statement. One of the key reasons why Sigmoid is not preferred over ReLU is due its optimization properties (vanishing gradients). Since the EAFO framework is completely agnostic to that, the contribution of this framework becomes significantly weaker. If the authors could empirically show how EAFO could be used in practice or justify the choice of entropy as an indicator for activation function optimality, that could help address my concerns
2. Another suggestion is to actually compare the entropy post training of neural networks trained with different activations, not just at the end, but also in the intermediate layers. Since the activation function is being used throughout the network, does lower entropy also help there? If not, should only be the last few layers be equipped with CRReLU?

---

> ### Author Response · Authors · 2024-11-21
>
> Dear Reviewer YJtL:
>
> Thank you for your efforts on our work, for your insightful comments and for your constructive suggestions. Based on your comments, we summarize the
> strength of our work as follows:
>
> 1. The paper is written **clearly and concisely**, and is **easy to read**.
> 2. The proposed framework is a **great** approach to search for activation functions and could **potentially** generate insights. Furthermore, the proposed WAFBC serves as **a good stepping stone** for future follow up works.
>
> And the weaknesses as follows:
>
> 1. Further discussion on why a lower entropy indicates a better classification.
>
> 2. Rank several activation functions using the EAFO framework, with the statement of lower entropy indicating better classification performance.
>
> 3. Add error bars for the experiments in order to better understand statistical significance of the results.
>
> **W1 and Q1**:
>
> Thank you once more for your insightful comment and for your constructive suggestions. We would like to respond this question intuitively, empirically, and theoretically, that is, why a lower entropy indicates a better classification. From the **intuitive** perspective, lower entropy indicates less uncertainty for feature representation, which usually means more information is captured in fewer features. In other words, lower entropy can suggest that features are more discriminative, better able to distinguish different categories or patterns. From the
> **empirical** perspective, early work [1] experimentally showed that minimization of Shannon’s entropy of the gap between the output and the desired target could achieve a better performance compared to MSE and CE. In early work [2], the authors experimentally illustrated that minimizing entropy of the error between output and desired targets yields exceptionally satisfactory classification performance. From the **theoretical** perspective, work [3] proved that for training DNN classifiers essentially learns the conditional entropy of the underlying data distribution of the dataset (the information or uncertainty remained in the labels after revealing the input) and derived the mutual information (between the corresponding feature and the label) bounds for a classification data model (Section 7). Hence, the conditional entropy $H$(output|input) will decrease with the process of training. In the work [4], the authors derived upper bounds on the generalization error in terms of the mutual information between its input and output. According to [4], a smaller mutual information means a smaller generalization error upper bound, which in turn suggests better classification performance. We have mutual information $I$(input,output) = $H$(output) - $H$(output|input). with the process of training, $H$(output|input) decreases; hence, in order to make the mutual information $I$(input,output) as small as possible, we should minimize the $H$(output). Therefore, we consider that a lower entropy signifies better classification performance.
>
> Furthermore, you've noted that the prevalent belief is that ReLU outperforms Sigmoid due to its immunity to the vanishing gradients issue, which is indeed accurate. In our research, we merely consider this matter from a different point. In our work, discussion on them is delineated within **the WAFBC part** (Section 4.2, line 229-232), rather than the EAFO part (Section 4.3).
>
> [1]Silva L M, de Sá J M, Alexandre L A. Neural network classification using Shannon's entropy[C]//Esann. 2005: 217-222.
>
> [2]Santos J M, Alexandre L A, de Sá J M. The error entropy minimization algorithm for neural network classification[C]//int. conf. on recent advances in soft computing. 2004: 92-97.
>
> [3]Yi J, Zhang Q, Chen Z, et al. Mutual information learned classifiers: An information-theoretic viewpoint of training deep learning classification systems[J]. arXiv preprint arXiv:2209.10058, 2022.
>
> [4]Xu A, Raginsky M. Information-theoretic analysis of generalization capability of learning algorithms[J]. Advances in neural information processing systems, 2017, 30.

---

> ### Author Response · Authors · 2024-11-21
>
> **W2**:
>
> Thank you once more for your insightful comment and for your constructive suggestions. Actually, EAFO (fully known as Entropy-based Activation Function Optimization) aims at optimization instead of comparison. While, based on your suggestion, we would like to provide a little insight through comparison of entropy.
> The information entropy takes the form as (line 161):
> \\[
> H(y(x))=-\\int p(y(x))y'(x) \\log (p(y(x)y'(x))) dx
> \\]
> where $y(x)$ is the inverse function of the activation function.
>
> **Insight1** Under mild assumptions, PReLU with tunable parameters should outperform the Leaky-ReLU with fixed-parameters.
>
> proof of scratch:
>  f(x)=x(x$\\succ$0) and f(x)=$\\alpha$x(x$\\prec$0): for PReLU, $\\alpha$ is tunable ; while for Leaky-ReLU, $\\alpha$ is fixed. The inverse function takes the form as y(x)=x(x$\\succ$0) and y(x)=x/$\\alpha$(x$\\prec$0).
> We segregate the positive and negative components of the entropy function:
> \\[\begin{split}
> H(y(x))&=-\int_{-\infty}^{0} p(y(x))y'(x) \log (p(y(x)y'(x))) dx -\int_{0}^{+\infty} p(y(x))y'(x) \log (p(y(x)y'(x))) dx \\
> &=-\int_{-\infty}^{0} p(x/\alpha)/\alpha \cdot \log (p(x/\alpha)/\alpha) dx -\int_{0}^{+\infty} p(x)\log (p(x)) dx
> \end{split}
> \\]
>
> Hence,
> \\[
> H(\text{PReLU})-H(\text{Leaky-ReLU}) = -\int_{-\infty}^{0} p(x/\alpha_1)/\alpha_1 \cdot \log (p(x/\alpha_1)/\alpha_1)- p(x/\alpha_2)/\alpha_2 \cdot \log (p(x/\alpha_2)/\alpha_2)dx
> \\]
> where $\\alpha_1$ represents tunable parameter of PReLU; $\\alpha_2$ represents fixed parameter of Leaky-ReLU.
> Moreover, from the formula, due to the PReLU's ability to dynamically adjust its parameters based on the data distribution $p(\\cdot)$, the resulting mutual information will be lower compared to the Leaky-ReLU with fixed parameters, resulting in better classification performance.
>
> While, it is crucial to recognize that such a statement is not strictly accurate. Alteration of parameter $\alpha$ in response to the data distribution will undoubtedly vary across different network architectures. Moreover, we would like to say that it appears to be a rather challenging task to rank different activation functions in a generalized condtion (the ranking needs a lot strong assumption); for different network architectures, initialization and stochasticity, the theoretical understanding of this issue still requires a considerable amount of discussion and comprehension.
>
> **W3**
>
> We sincerely appreciate the constructive suggestion, as it has greatly help us refine the paper and further enhance the results. The results from the original article train only one run. Following your suggestion, we conducted three runs on all results to gain a deeper understanding of the statistical significance, reporting both the mean and standard deviation. Experiments on CIFAR10 and CIFAR100 are conducted on 4 RTX3090, and additional experiments on ImageNet1K are conducted on 4 NVIDIA L20.
>
> Table 1: Test accuracy of experiments conducted on CIFAR10 for 100 epochs with error bar.
>
> |      | GELU               | ELU                | PReLU              | CELU               | SiLU               | Mish               | CRReLU             |
> |:----:|:------------------:|:------------------:|:------------------:|:------------------:|:------------------:|:------------------:|:------------------:|
> | ViT  | 0\.704$\\pm$0\.002 | 0\.664$\\pm$0\.005 | 0\.780$\\pm$0\.006 | 0\.665$\\pm$0\.006 | 0\.686$\\pm$0\.003 | 0\.687$\\pm$0\.003 | **0\.807$\\pm$0\.003** |
> | DeiT | 0\.724$\\pm$0\.007 | 0\.676$\\pm$0\.006 | 0\.754$\\pm$0\.001 | 0\.677$\\pm$0\.008 | 0\.699$\\pm$0\.005 | 0\.702$\\pm$0\.006 | **0\.770$\\pm$0\.003** |
> | TNT  | 0\.737$\\pm$0\.005 | 0\.695$\\pm$0\.006 | 0\.758$\\pm$0\.003 | 0\.687$\\pm$0\.002 | 0\.711$\\pm$0\.007 | 0\.716$\\pm$0\.008 | **0\.769$\\pm$0\.005** |
>
> Table 2: Test accuracy of experiments conducted on CIFAR100 for 100 epochs with error bar.
> |      | GELU               | ELU                | PReLU              | CELU               | SiLU               | Mish               | CRReLU             |
> |:----:|:------------------:|:------------------:|:------------------:|:------------------:|:------------------:|:------------------:|:------------------:|
> | ViT  | 0\.326$\\pm$0\.008 | 0\.289$\\pm$0\.001 | 0\.432$\\pm$0\.010 | 0\.289$\\pm$0\.002 | 0\.312$\\pm$0\.006 | 0\.306$\\pm$0\.008 | **0\.466$\\pm$0\.006** |
> | DeiT | 0\.466$\\pm$0\.009 | 0\.405$\\pm$0\.005 | 0\.500$\\pm$0\.005 | 0\.405$\\pm$0\.005 | 0\.435$\\pm$0\.006 | 0\.438$\\pm$0\.010 | **0\.507$\\pm$0\.001** |
> | TNT  | 0\.475$\\pm$0\.008 | 0\.436$\\pm$0\.003 | 0\.490$\\pm$0\.007 | 0\.430$\\pm$0\.005 | 0\.450$\\pm$0\.009 | 0\.455$\\pm$0\.008 | **0\.509$\\pm$0\.004** |

---

> ### Author Response · Authors · 2024-11-21
>
> Table3: Test accuracy of experiments conducted on ImageNet1K for 100 epochs with error bar.
> |      | GELU               | ELU                | PReLU              | CELU               | SiLU               | Mish               | CRReLU             |
> |:----:|:------------------:|:------------------:|:------------------:|:------------------:|:------------------:|:------------------:|:------------------:|
> | ViT  | 0\.539$\\pm$0\.003 | 0\.372$\\pm$0\.006 | 0\.568$\\pm$0\.004 | 0\.376$\\pm$0\.005 | 0\.461$\\pm$0\.007 | 0\.469$\\pm$0\.011 | **0\.575$\\pm$0\.004** |
> | DeiT | **0\.617$\\pm$0\.004** | 0\.491$\\pm$0\.007 | 0\.608$\\pm$0\.004 | 0\.489$\\pm$0\.008 | 0\.585$\\pm$0\.007 | 0\.589$\\pm$0\.003 | **0\.616$\\pm$0\.002** |
>
> **Q3**
>
> Thank you once more for your insightful comment and for your constructive suggestions. We consider this suggestion to be exceedingly valuable and remarkably insightful. Following your suggestions, we apply the post-trained ViT-Tiny with CRReLU and GELU on ImageNet1K. By randomly selecting same ten batches of images from ImageNet, we compute the information entropy after each of the 12 layers. We present the mean and standard deviation of these values as follows:
>
> Table4: Entropy calculation after activation (GELU and CRReLU) on 12 layers of the trained ViT on ImageNet1K.
> | Layer  | 1                  | 2                  | 3                  | 4                  | 5                  | 6                  |
> |:------:|:------------------:|:------------------:|:------------------:|:------------------:|:------------------:|:------------------:|
> | CRReLU | 7\.594$\\pm$0\.007 | 7\.598$\\pm$0\.003 | 7\.599$\\pm$0\.003 | 7\.595$\\pm$0\.003 | 7\.592$\\pm$0\.003 | 7\.584$\\pm$0\.004 |
> | GELU   | 7\.536$\\pm$0\.046 | 7\.541$\\pm$0\.019 | 7\.561$\\pm$0\.011 | 7\.573$\\pm$0\.006 | 7\.580$\\pm$0\.005 | 7\.583$\\pm$0\.004 |
>
> | Layer  | 7                  | 8                  | 9                  | 10                 | 11                 | 12                 |
> |:------:|:------------------:|:------------------:|:------------------:|:------------------:|:------------------:|:------------------:|
> | CRReLU | 7\.572$\\pm$0\.005 | 7\.557$\\pm$0\.005 | 7\.540$\\pm$0\.005 | 7\.523$\\pm$0\.007 | 7\.498$\\pm$0\.008 | 7\.461$\\pm$0\.008 |
> | GELU   | 7\.585$\\pm$0\.004 | 7\.585$\\pm$0\.004 | 7\.583$\\pm$0\.004 | 7\.580$\\pm$0\.004 | 7\.577$\\pm$0\.004 | 7\.560$\\pm$0\.004 |
>
> From the results presented above, it is evident that for GELU, the entropy after 12 layers of activation exhibits an overall increasing trend, whereas conversely, CRReLU demonstrates a general declining trend. Furthermore, we have noted that the reduction in entropy for CRReLU between layers 1 and 6 is not significant, whereas a marked decline is observed from layers 7 to 12. In light of your suggestion, we employ GELU for layers 1 to 6 and CRReLU for layers 7 to 12, denoting this as "6GELU+6CRReLU".  We conduct three runs on CIFAR10, CIFAR100, and ImageNet1K, presenting the mean and standard deviation of the results as follows. Experiments on CIFAR10 and CIFAR100 are conducted on 4 RTX3090, and those on ImageNet are carried out on 4 NVIDIA L20.
>
> Table5: Test accuracy of experiments conducted with ViT (12GELU, 6GELU+6CRReLU, 12CRReLU) for 100 epochs with error bar
>
> |            | 12GELU             | 6GELU\+6CRReLU     | 12CRReLU           |
> |:----------:|:------------------:|:------------------:|:------------------:|
> | CIFAR10    | 0\.704$\\pm$0\.002 | 0\.755$\\pm$0\.008 | 0\.807$\\pm$0\.003 |
> | CIFAR100   | 0\.326$\\pm$0\.008 | 0\.399$\\pm$0\.004 | 0\.466$\\pm$0\.006 |
> | ImageNet1K | 0\.539$\\pm$0\.003 | 0\.512$\\pm$0\.001 | 0\.575$\\pm$0\.004 |
>
> From the results, it appears that having only the last few layers equipped with CRReLU is not as effective as utilizing CRReLU throughout the entire network. Especially the results on ImageNet1K, 6GELU+6CRReLU is significantly and stably worsen to all GELU and all CRReLU, which is quite surprising to us. We consider that this may be due to the fact that, while the reduction in entropy is not significantly apparent in the earlier layers, CRReLU's focus on achieving lower entropy still facilitates superior feature extraction. It seems that when using GELU in the earlier layers and CRReLU in the later layers, on small-scale datasets, it is still possible to benefit from the CRReLU mechanism in the later layers (the features learned in the earlier layers are not good enough yet); however, on large-scale datasets, the features learned in the earlier layers might even have a negative effect.
>
> Finally, we would like to thank you once again for the constructive suggestions and insightful comments on our work.
>
> Warm regards,
>
> Authors of submission 6110

---

> ### Author Response · Authors · 2024-11-29
> **Sincerely Seeking Your Invaluable Feedback**
>
> Dear Reviewer YJtL:
>
> We hope this message finds you well. As the discussion period draws to a close, we are reaching out to solicit your thoughts on the rebuttal responses and the revised manuscript, inspired by your valuable insights. We have provided additional supportive experiments and conducted further discussions in the rebuttal responses and the revised manuscript.
>
> We would like to briefly summarize the changes we made to the manuscript for your easier navigation. On the additional supportive experiments, specifically, we focus on the following aspects: enhancing all experimental results with three runs, additional architecture (Appendix F.1), additional dataset (Appendix F.2), additional $\epsilon$ initialization (Appendix F.3), entropy calculation after activation (Appendix F.4) and mixed activation function (Appendix F.5). On the additional discussions, we focus on Lipschitz continuity analysis (Appendix G), initialization and training stability (Appendix H), lower entropy indicates better classification (Appendix I) and dynamic optimization (Appendix J).
>
> Your expertise in this domain has been a guiding light in these improvements, and we deeply appreciate your constructive and insightful comments. If there are any remaining questions or concerns, we would be more than happy to discuss further. Could you kindly let us know if the points we addressed resolve your concerns, and if you would consider revisiting your evaluation score based on the additional contents?
>
> Thank you once again for your thoughtful feedback and engagement, as it has greatly contributed to improving the quality of our work.
>
> Warm regards,
>
> Authors of Submission 6110

---

> > ### Author Response · Authors · 2024-12-02
> > **Sincerely Seeking Your Invaluable Feedback**
> >
> > Dear Reviewer YJtL:
> >
> > We hope this message finds you well. As the discussion period draws to a close in 20 hours, we are reaching out to solicit your thoughts on the rebuttal responses and the revised manuscript, inspired by your valuable insights. We have provided additional supportive experiments and conducted further discussions in the rebuttal responses and the revised manuscript.
> >
> > Your feedback is invaluable, and we deeply appreciate your time and effort. If there are any remaining questions or concerns, we would be more than happy to clarify further. Could you kindly let us know if the points we addressed resolve your concerns, and if you would consider revisiting your evaluation score based on the additional evidence?
> >
> > Best regards,
> >
> > Authors of Submission 6110

---

> > > ### Author Response · Authors · 2024-12-03
> > > **Discussion Period draws to a Close in 8 Hours. We are Sincerely Seeking Your Invaluable Feedback.**
> > >
> > > Dear Reviewer YJtL:
> > >
> > > We hope this message finds you well. As the discussion period draws to a close in less than 8 hours, we are reaching out to solicit your thoughts on the rebuttal responses, the revised manuscript and the latest version of the paper (Please download it at the following anonymous link https://anonymous.4open.science/r/Revised_Paper-ICLR2025_6110_submission/ICLR_2025_6110_submission.pdf ). In the latest version, we have:
> > >
> > > 1.	**incorporate all the additional experiments and discussions in the rebuttal**.
> > >
> > > 2.	transition to **a research question-oriented** presentation style, demonstrating a significant improvement in both **content organization** and **presentation clarity**.
> > >
> > > 3.	**balance the length** of the main text and the appendix.
> > >
> > > The following is a more specific elaboration of these modifications.
> > >
> > > In the main text:
> > >
> > > 1.	In the **Introduction**, we present the **three questions** on which the content of this paper is based. And in the **summary part** of Introduction, we show the work we have done to **answer** these three questions.
> > >
> > > 2.	In **Section 4.2**, we give the answer to **Question 1**.
> > >
> > > 3.	In **Section 4.3**, we give the answer to **Question 2**. We change the **presentation form of EAFO methodology outline** in a more aesthetically pleasing manner.
> > >
> > > 4.	In **Section 4.4**, we give the answer to **Question 3**. We add the **main conclusions of Lipschitz Continuity Analysis**.
> > >
> > > 5.	In **Section 5.1**, we add **main results** of **Entropy Analysis across Network Layers**. We **briefly** mentioned **Additional Experiments on Architecture and Dataset**.
> > >
> > > 6.	In the **Discussion**, we provide further discussion on **potential applications**.
> > >
> > > 7.	During writing process, we **cite all content in appendix** in order to facilitate a **good correspondence** for readers.
> > >
> > > In the appendix:
> > >
> > > 1.	We provide detailed **proof of Lipschitz Continuity Analysis** in Appendix E.
> > >
> > > 2.	We provide **additional experiments on architecture and dataset** in Appendix F.
> > >
> > > 3.	We provide **additional experiments on mixed activation function** in Appendix G.
> > >
> > > 4.	We provide **further discussion on initialization and training stability** in Appendix H.
> > >
> > > 5.	We provide **further discussion on lower entropy indicates better classification** in Appendix I.
> > >
> > > 6.	We provide **further discussion on bias towards activation function in pre-trained models** in Appendix J.
> > >
> > > 7.	We provide **further discussion on dynamic optimization** in Appendix K.
> > >
> > > 8.	We provide **further discussion on activation function ranking** in Appendix L.
> > >
> > > 9.	We provide **further discussion on LLM inference task** in Appendix M.
> > >
> > > Your feedback is invaluable, and we deeply appreciate your time and effort. If there are any remaining questions or concerns, we would be more than happy to clarify further. Could you kindly let us know if the points we addressed resolve your concerns, and if you would consider revisiting your evaluation score based on the additional evidence?
> > >
> > > Best regards,
> > >
> > > Authors of Submission 6110

---

> ### Comment · Reviewer_YJtL · 2024-12-03
> **Response to authors**
>
> Thank you authors for a detailed response. In light of the new experimental results and new discussion, I am bumping my rating from 3 to 5. I think all this discussion added by authors in the paper make it a much stronger submission than before. However the wider utility of the proposed framework is the reason for not further increasing my rating.

---

> ### Author Response · Authors · 2024-12-03
> **Thank you for the Feedback. We are pleased to provide Further Clarification.**
>
> Dear Reviewer YJtL：
>
> We are more than happy to receive your invaluable feedback. Based on your feedback, we understand that you believe our paper is now a much stronger submission than before. Therefore, we are confident that we have addressed all of (or at least a majority of) your concerns.
>
> Concurrently, we also noticed that you still have doubts about the wider utility of the proposed framework. We are pleased to provide further clarification on this matter. The Entropy-based Activation Function Optimization (EAFO) framework mainly aims at “Optimization”. *Firstly*, the framework will not be limited only to invertible activation functions. The potential approach to addressing other non-invertible activation functions involves using the Lebesgue integral form instead of the original Riemann integral used in the entropy calculation. *Secondly*, the framework shows potential to implement activation function iteration optimization during neural network training and has laid a solid foundation for the community addressing such open challenge in the future. *Moreover*, this framework also provides insights for novel networks that are focused on activation optimization, such as KANs. *Finally*, we believe this framework can still provide insights on your mentioned “Activation Function Ranking”, as shown in the rebuttal response. We believe that you raised this question due to the needs of your other research, and that you will have a more profound understanding of this problem. Although we are not yet able to solve this problem perfectly in such a short time, we believe further discussions between us will provide more insights to it and we look forward to further discussions with you.
>
> Finally, we would like to thank you once more for your invaluable feedback.
>
> Best regards,
>
> Authors of Submission 6110

---

### Author Response · Authors · 2024-11-24
**Sincerely Looking Forward to Discussion**

Dear reviewers:

We would like to thank you once again for your great efforts on our work, for your insightful comments and for your constructive suggestions. As the discussion process only left less than 72 hours, we are reaching out to solicit your thoughts on the rebuttal responses. We have gone through your points one-by-one and tried to address them carefully. We eagerly anticipate your feedback and hope our efforts align with the needs of the community and the rigour of the conference.

Warm regards,

Authors of submission 6110

---

### Author Response · Authors · 2024-11-28
**Revisions on the Manuscript**

Dear reviewers:

We would like to thank you once again for your great efforts and time on our work, for your insightful comments and for your constructive suggestions. We were glad to see that reviewers have highlighted advantages of this paper, in that the proposed methodology and activation function is **novel** (Reviewer LUV6, Reviewer m9pv) and serves as a **good stepping stone** for future follow up works (Reviewer YJtL), the theoretical foundation is **solid** and **rigorous** (Reviewer LUV6, Reviewer m9pv), the empirically evaluation is **comprehensive** (Reviewer m9pv, Reviewer LUV6) and **thorough** (Reviewer LUV6)，the performance is **significantly better than** the SOTA (Reviewer 7aze), presentation of the paper is **clear and concise** (Reviewer YJtL, Reviewer LUV6).

We have carefully considered your precious feedback and have made revisions to the manuscript to address your concerns. All additions are colored in $\color{blue}{blue}$ for your easier review. Below, we would like to outline the key changes implemented.

## Additional Experiments

1.	To strengthen the experimental results, we further conduct all experiments three runs (Table1, Table2, Table3, Table4).
2.	Experiments on generalization to network architecture (Appendix F.1).
3.	Experiments of performance on additional dataset (Appendix F.2).
4.	Experiments on exploring the impact of different initial values of $\epsilon$, as well as potential instabilities or failure cases under different initialization schemes (Appendix F.3).
5.	Entropy of post-trained neural networks’ comparison (Appendix F.4).
6.	Experiments of performance comparison with mixed activation function (Appendix F.5).

## Additional Discussion

1.	Lipschitz continuity analysis (Appendix G). Lipschitz continuity constitutes a stronger form of continuity, which imposes an upper bound on the rate of variation of a function. We calculate the Lipschitz constants for (GELU,) SiLU, Mish, and CRReLU; furthermore, we derive the recommended $\epsilon$ initialization range.
2.	Further discussion on initialization and training stability (Appendix H).
3.	Further discussion on lower entropy indicates better classification (Appendix I).
4.	Further discussion on dynamic optimization (Appendix J).

We hope these manuscript revisions could address your concerns effectively. Your precious feedback has been instrumental in improving our work, and we thank you again for your constructive input.

Best regards,

Authors of Submission 6110

---

> ### Comment · Reviewer_LUV6 · 2024-11-30
> **Suggestions from Reviewer LUV6 to the Authors**
>
> Dear Authors,
>
> I have thoroughly reviewed the authors' responses and my fellow reviewers' comments. Please refer to my detailed response for specifics. Herein, I strongly recommend the authors to incorporate these valuable discussions and experimental results into the revised manuscript, as they significantly strengthen both the theoretical foundations and empirical validations of this work. To further enhance the manuscript's impact, I suggest:
>
> - Consider adding a concise summary of the Lipschitz continuity analysis in the main text, as this provides crucial theoretical grounding for the initialization strategy.
> - Include key findings from the entropy analysis across network layers in the primary results section, as this offers important insights into the method's behavior.
> - Consider expanding the discussion on potential applications and limitations in more complex network architectures.
>
> I hope my suggestions help to further strengthen this paper. I also look forward to further discussions.
>
> Best regards,
>
> Reviewer LUV6

---

> > ### Author Response · Authors · 2024-11-30
> > **Thank you for Precious Suggestions**
> >
> > Dear Reviewer LUV6:
> >
> > We would like to thank you once again for your invaluable feedback and precious suggestions on the manuscript. We will incorporate **all the additional experiments and discussions in the rebuttal** in a more coherent and logical framework in the next version. And we will incorporate **all of your precious suggestions** into the next version of the paper; they are indeed profoundly beneficial.
> >
> > Best regards,
> >
> > Authors of Submission 6110

---

> > > ### Comment · Reviewer_LUV6 · 2024-11-30
> > > **Additional Comments by Reviewer LUV6 to Authors**
> > >
> > > Dear Authors,
> > >
> > > Thank you for your kind acknowledgment. I appreciate that my suggestions can help strengthen this work further. In particular, I am pleased with your commitment to balancing technical depth with broad accessibility - this is crucial for maximizing the paper's impact on the field. Note that I have outlined several additional recommendations in my latest response that I believe could enhance the manuscript's clarity. Please check it out. I remain actively engaged in this review process and encourage you to reach out if you would like any clarification regarding my previous suggestions.
> > >
> > > Additionally, I hope these comments help my fellow reviewers and Area Chairs better understand the basis of my recommendation.
> > >
> > > Best regards,
> > >
> > > Reviewer LUV6

---

> ### Author Response · Authors · 2024-12-02
> **Thank you for the Precious Suggestions. We have completed Next Version of the Paper.**
>
> Dear Reviewer LUV6:
>
> We would like to thank you once more for your invaluable feedback and precious suggestions on the manuscript. Based on your suggestions, we have completed the next version of the paper. Please download the latest version of the paper at https://anonymous.4open.science/r/Revised_Paper-ICLR2025_6110_submission/ICLR_2025_6110_submission.pdf. In this version, we make some changes to the **paper's presentation style** and **adjust the text colors to some extent**, so we **do not** use color to indicate modified content with the purpose of avoiding color confusion. In this version of paper, we:
>
> 1.	**incorporate all the additional experiments and discussions in the rebuttal**
> 2.	transition **presentation style** from a largely method-oriented way to a **research question-oriented**
> 3.	**balance the length** of the main text and the appendix
>
> The following is a more specific elaboration of these modifications.
>
> In the main text:
>
> 1.	In the **Introduction**, we present the **three questions** on which the content of this paper is based. And in the **summary part** of Introduction, we show the work we have done to **answer** these three questions.
> 2.	In **Section 4.2**, we give the answer to **Question 1**.
> 3.	In **Section 4.3**, we give the answer to **Question 2**. We change the **presentation form of EAFO methodology outline** in a **more aesthetically pleasing** manner.
> 4.	In **Section 4.4**, we give the answer to **Question 3**. We add the **main conclusions of Lipschitz Continuity Analysis**.
> 5.	In **Section 5.1**, we add **main results** of **Entropy Analysis across Network Layers**. We **briefly** mentioned **Additional Experiments on Architecture and Dataset**.
> 6.	In the **Discussion**, we provide further discussion on **potential applications**.
> 7.	During writing process, we **cite all content in appendix** in order to facilitate a **good correspondence** for readers.
>
> In the appendix:
>
> 1.	We provide detailed **proof of Lipschitz Continuity Analysis** in Appendix E.
> 2.	We provide **additional experiments on architecture and dataset** in Appendix F.
> 3.	We provide **additional experiments on mixed activation function** in Appendix G.
> 4.	We provide **further discussion on initialization and training stability** in Appendix H.
> 5.	We provide **further discussion on lower entropy indicates better classification** in Appendix I.
> 6.	We provide **further discussion on bias towards activation function in pre-trained models** in Appendix J.
> 7.	We provide **further discussion on dynamic optimization** in Appendix K.
> 8.	We provide **further discussion on activation function ranking** in Appendix L.
> 9.	We provide **further discussion on LLM inference task** in Appendix M.
>
> In the next version, we will continue to strengthen the connections between sections and polish the language of our paper.
>
> Finally, we would like to thank you once again for your invaluable feedback and precious suggestions on the manuscript.
>
> Best regards,
>
> Authors of Submission 6110

---

> ### Comment · Reviewer_LUV6 · 2024-12-03
> **Official Response by Reviewer LUV6 to the Authors' Latest Revision**
>
> Dear Authors,
>
> I have thoroughly reviewed your latest response and the revised manuscript with the changes you've outlined. I am pleased to see the comprehensiveness of the revisions, particularly the transition to a research question-oriented presentation style, which demonstrates a significant improvement in both **content organization** and **presentation clarity**. The enhanced logical flow effectively conveys the key research questions and contributions of this work to the field.
>
> The restructuring of the paper around fundamental questions in the Introduction, with corresponding answers developed through Sections 4.2-4.4, significantly enhances the paper's logical flow and accessibility. The addition of **Lipschitz Continuity Analysis** in Section 4.4 and the **Entropy Analysis across Network Layers** in Section 5.1 addresses key concerns I had raised previously, providing robust foundations for the methodology.
>
> I am particularly impressed with the revised appendix, which now provide **detailed supporting evidence** while maintaining excellent readability in the main text. The additional discussions on initialization, training stability, and practical applications demonstrate both theoretical rigor and practical applicability of the proposed method.
>
> Given these improvements, I am revising my rating from 6 to 8, as I suppose this may better reflect the current overall quality of this paper. I also encourage my fellow reviewers to re-examine the latest revised manuscript through the anonymous link. I remain available for further discussions that may benefit either the authors or other reviewers.
>
> Best regards,
>
> Reviewer LUV6

---

### Author Response · Authors · 2024-12-03
**Summary of Rebuttal and Discussion (Part 1)**

Dear Area Chair and reviewers:

We would like to thank you once again for your great efforts and time on our work, for your insightful comments and for your constructive suggestions. Below is a concise summary of the rebuttal and the discussion for ease of reference.
***
***Reviewer Highlights in the Original Review***

The paper has been recognized for its **solid and rigorous theoretical foundation**, **comprehensive and thorough empirical evaluation**,  **clear and concise presentation**. Key highlights include:

1.	The proposed methodology and activation function is **novel** (Reviewer LUV6, Reviewer m9pv) and **serves as a good stepping stone** for future follow up works (Reviewer YJtL) .
2.	The theoretical foundation is **solid** and **rigorous** (Reviewer LUV6, Reviewer m9pv).
3.	The empirical evaluation is **comprehensive** (Reviewer m9pv, Reviewer LUV6) and **thorough** (Reviewer LUV6).
4.	The performance of the novel activation function is **significantly better** than the SOTA (Reviewer 7aze).
5.	The presentation of the paper is **clear and concise** (Reviewer YJtL, Reviewer LUV6).
***
***Weaknesses that have been Addressed***

Reviewers raised concerns regarding:
* More comprehensive experimental verification (multiple experimental runs; more architecture and dataset; more initial values; entropy analysis across network layers; and mixed activation function verification)
* Clarification on existing experimental results (why paper results lower than SOTAs' paper reports; knowledge distillation bias toward activation functions)
* Framework clarification (why lower entropy indicates better classification)
* Further analysis on the novel activation function (convergence properties; recommended range of the initialization value; training stability; insights within other NLP tasks)
* Potential applications (dynamic optimization during iterative training, activation function ranking)

In the Rebuttal responses, we have **gone through the points one-by-one and addressed them carefully**. **Based on the reviewers' feedback, we believe that we have thoroughly addressed their concerns.**
***
***Reviewers’ Feedback about the Rebuttal and the Initial Revised Manuscript***

In the discussion period, Reviewer LUV6 has offered perceptive and profound additional insights into improvements incorporated in the rebuttal and the presentation of the paper. We sincerely appreciate his precious suggestions and comprehensive participation during the discussion phase. From the feedback of Reviewer LUV6, it is considered that:
* we have made **substantial improvements** that address the key weaknesses.
* theoretical foundation has been **significantly strengthened**.
* we have provided **thoughtful responses** regarding the dynamic optimization challenges and it **offer viable paths forward**.
* the work now presents **a more complete contribution to the field**.

Furthermore, we are more than happy to hear from the feedback of Reviewer 7aze and Reviewer YJtL in the last day of the discussion phase. It is shown that Reviewer 7aze is **satisfied** with the answers and Reviewer YJtL considers the paper **a much stronger submission** than before. We appreciate their feedback.
***

---

> ### Author Response · Authors · 2024-12-03
> **Summary of Rebuttal and Discussion (Part 2)**
>
> ***
> ***Latest Version of the Paper***
>
> In order to show **presentation for next version of the paper** more clearly, we provide the latest version of the paper with an anonymous link. **Please download at** https://anonymous.4open.science/r/Revised_Paper-ICLR2025_6110_submission/ICLR_2025_6110_submission.pdf (We will make this link active until Jan, 23, 2025 AOE and will not update it after Dec, 2, 2024 AOE).  The following is a more specific elaboration of our modifications.
>
> In the main text:
>
> 1.	In the **Introduction**, we present the **three questions** on which the content of this paper is based. And in the **summary part** of Introduction, we show the work we have done to **answer** these three questions.
> 2.	In **Section 4.2**, we give the answer to **Question 1**.
> 3.	In **Section 4.3**, we give the answer to **Question 2**. We change the **presentation form of EAFO methodology outline** in a more aesthetically pleasing manner.
> 4.	In **Section 4.4**, we give the answer to **Question 3**. We add the **main conclusions of Lipschitz Continuity Analysis**.
> 5.	In **Section 5.1**, we add **main results** of **Entropy Analysis across Network Layers**. We **briefly** mentioned **Additional Experiments on Architecture and Dataset**.
> 6.	In the **Discussion**, we provide further discussion on **potential applications**.
> 7.	During writing process, we **cite all content in appendix** in order to facilitate a **good correspondence** for readers.
>
> In the appendix:
>
> 1.	We provide detailed **proof of Lipschitz Continuity Analysis** in Appendix E.
> 2.	We provide **additional experiments on architecture and dataset** in Appendix F.
> 3.	We provide **additional experiments on mixed activation function** in Appendix G.
> 4.	We provide **further discussion on initialization and training stability** in Appendix H.
> 5.	We provide **further discussion on lower entropy indicates better classification** in Appendix I.
> 6.	We provide **further discussion on bias towards activation function in pre-trained models** in Appendix J.
> 7.	We provide **further discussion on dynamic optimization** in Appendix K.
> 8.	We provide **further discussion on activation function ranking** in Appendix L.
> 9.	We provide **further discussion on LLM inference task** in Appendix M.
>
> In the next version, we will continue to **strengthen the connections between sections**, **polish the language** of our paper and also **ensure its format complies with requirements of ICLR 2025** (especially the limitation of 10 pages in the main text).
>
> ***
>
> ***Reviewer’s Feedback on the Latest Version of the Paper***
>
> Reviewer LUV6 have thoroughly reviewed our latest version of the paper. From the feedback, it is considered that:
>
> * the revisions present **comprehensiveness**; the transition to a research question-oriented presentation style demonstrating a **significant improvement** in both **content organization** and **presentation clarity**.
> * the restructuring of the paper **significantly enhances the paper's logical flow and accessibility**.
> * the addition of Lipschitz Continuity Analysis and the Entropy Analysis across Network Layers provides **robust foundations** for the methodology.
> * the revised appendix provide **detailed supporting evidence** while **maintaining excellent readability in the main text**.
> * the additional discussions demonstrate **both theoretical rigor and practical applicability** of the proposed method.
> * overall, the work can provide **knowledge advancement to the field**.
> ***
>
> Finally, we would like to thank you once more for your great efforts and time on our work, for your insightful comments and for your constructive suggestions.
>
> Best regards,
>
> Authors of Submission 6110

---

### Meta-Review · Area_Chair_APb5 · 2024-12-24

**Metareview:**

This paper proposes an entropy-based activation function optimization method from the perspective of information entropy, and derives a new activation function called Corrected Regularized ReLU (CRReLU). The presentation of the paper is clear, the proposed method is novel, and its performance significantly outperforms the state-of-the-art (SoTA). Some reviewers raised concerns about the experimental results and the analysis of the new activation function. The authors addressed most of these issues during the defense and provided detailed analysis. Overall, this is a paper with a rich theoretical foundation and comprehensive experimental validation, so the AC recommends accept.

**Additional Comments On Reviewer Discussion:**

The main concerns raised by the reviewers were the lack of more comprehensive experimental validation and the need for analysis and clarification of the existing experimental results. The authors added many experiments and carefully analyzed them, addressing most of the reviewers' questions.

---

### Decision · Program_Chairs · 2025-01-22

Accept (Poster)